# Coordinated regulation of chemotaxis and resistance to copper by CsoR in *Pseudomonas putida*

Meina He, Yongxin Tao, Kexin Mu, Haoqi Feng, Ying Fan, Tong Liu, Qiaoyun Huang, Yujie Xiao*, Wenli Chen*

National Key Laboratory of Agricultural Microbiology, Huazhong Agricultural University, Wuhan, China

## eLife Assessment

Data presented in this **useful** report suggest a potentially new model for chemotaxis regulation in the gram-negative bacterium P. putida. Data supporting interactions between CheA and the copper-binding protein CsoR, reveal potential mechanisms for coordinating chemotaxis and copper resistance. There was, however, concern about the large number of CheA interactors identified in the initial screen and it was felt that the study was **incomplete** without a substantial number of additional experiments to test the model and bolster the authors' conclusions.

**\*For correspondence:**
yjxiao@mail.hzau.edu.cn (YX);
wlchen@mail.hzau.edu.cn (WC)

**Competing interest:** The authors declare that no competing interests exist.

**Abstract** Copper is an essential enzyme cofactor in bacteria, but excess copper is highly toxic. Bacteria can cope with copper stress by increasing copper resistance and initiating chemorepellent response. However, it remains unclear how bacteria coordinate chemotaxis and resistance to copper. By screening proteins that interacted with the chemotaxis kinase CheA, we identified a copper-binding repressor CsoR that interacted with CheA in *Pseudomonas putida*. CsoR interacted with the HPT (P1), Dimer (P3), and HATPase_c (P4) domains of CheA and inhibited CheA autophosphorylation, resulting in decreased chemotaxis. The copper-binding of CsoR weakened its interaction with CheA, which relieved the inhibition of chemotaxis by CsoR. In addition, CsoR bound to the promoter of copper-resistance genes to inhibit gene expression, and copper-binding released CsoR from the promoter, leading to increased gene expression and copper resistance. *P. putida* cells exhibited a chemorepellent response to copper in a CheA-dependent manner, and CsoR inhibited the chemorepellent response to copper. Besides, the CheA-CsoR interaction also existed in proteins from several other bacterial species. Our results revealed a mechanism by which bacteria coordinately regulated chemotaxis and resistance to copper by CsoR.

## Introduction

Chemotaxis is a widespread ability of motile bacteria to direct their movement towards higher concentrations of beneficial chemicals or lower concentrations of toxic chemicals (*Keegstra et al., 2022*; *Wadhams and Armitage, 2004*). Chemotaxis plays a vital role in bacterial exploration and adaptation to complex environments (*Zboralski and Filion, 2020*; *Boin et al., 2004*). The chemotaxis signaling pathway is extensively studied in the enteric bacteria *Escherichia coli* and *Salmonella enterica* serovar Typhimurium (*Wadhams and Armitage, 2004*; *Porter et al., 2011*). In *E. coli*, the chemotaxis system consists of five methyl-accepting chemotaxis proteins (MCPs) and six core components (the kinase CheA, the response regulator CheY, the coupling protein CheW, the methylesterase CheB, the phosphatase CheZ, and the methyltransferase CheR). In response to decreased attractant

or increased repellent, methylated MCPs and the coupling protein CheW activate CheA autophosphorylation (*Parkinson et al., 2015*; *Ortega et al., 2017*). The phosphorylated CheA (CheA-P) is a phosphodonor for the response regulator CheY and the methylesterase CheB. After CheY accepts the phosphate group from CheA, it binds FliM and FliN of the motor-switch complex, resulting in a switch in the flagellar rotational direction and bacteria tumbling (*Welch et al., 1993*). Meanwhile, on a slower timescale, phosphorylated CheB demethylates the active MCPs, thus reducing their ability to activate CheA, leading to decreased levels of phosphorylated CheY and less tumbling (*Stewart, 1993*). The phosphatase CheZ contributes to signal termination by removing the phosphoryl group from phosphorylated CheY, and the methyltransferase CheR contributes to signal adaptation by catalyzing the methylation of MCPs (*Wadhams and Armitage, 2004*; *Porter et al., 2011*).

Among the bacterial chemotaxis system components, the kinase CheA is a five-domain enzyme central to the chemotaxis signaling pathway (*Bilwes et al., 1999*; *Muok et al., 2020*). The five domains (P1-P5) of CheA each have distinct functions. The P1 domain (HPT domain) contains the phosphoryl-accepting histidine that becomes phosphorylated, the P2 domain (CheY-binding domain) docks the response regulator proteins CheY and CheB, the P3 domain (Dimer domain) dimerizes the CheA protein, the P4 domain (HATPase_c domain) binds ATP and catalyzes phosphoryl transfer to the histidine residue on P1, and the P5 domain (CheW-binding domain) couples CheA to other chemotaxis components by binding both CheW and the chemoreceptors (*Bilwes et al., 1999*; *Muok et al., 2020*). Except for interacting with the components in chemotaxis system, CheA is reported to interact with proteins from other systems. For example, in the plant pathogen *Xanthomonas oryzae* pv. *oryzicola*, CheA interacts with and phosphorylates the response regulator VemR to regulate bacterial virulence, motility, and EPS production (*Cai et al., 2022*). In *Vibrio parahaemolyticus*, a polarly localized protein ParP interacts with CheA and prevents its dissociation from chemotaxis signaling arrays, facilitating proper chemotaxis and accurate inheritance of these macromolecular chemotactic machines (*Ringgaard et al., 2014*). In *Comamonas testosteroni*, CheA interacts with and phosphorylates the response regulator FlmD, resulting in decreased biofilm formation (*Huang et al., 2019*). In *Azospirillum brasilense* with two chemotaxis signaling systems, the kinase CheA from each chemotaxis signaling system physically interacts with the CheY response regulator of another system (*O'Neal et al., 2019*). In *Pseudomonas aeruginosa*, CheA interacts with the phosphodiesterase DipA to regulate its subcellular localization and activity, leading to individual cell heterogeneity and motility behavior diversity in bacterial populations (*Kulasekara et al., 2013*). These studies suggest that the CheA-mediated crosstalk between chemotaxis and other systems coordinates complex behaviors in diverse bacteria.

In most living organisms, copper is an essential cofactor for enzymes involved in fundamental processes such as respiration and photosynthesis (*Kim et al., 2008*; *Tsang et al., 2021*). However, copper also has toxic effects on cells, and bacteria have several strategies to increase their resistance to copper (*Giachino and Waldron, 2020*; *Andrei et al., 2020*; *Hyre et al., 2021*). The direct bacterial response associated with copper resistance is highly conserved and generally involves (a) sensing of the increased copper concentration by sensors, (b) activation of bespoke transcriptional networks, (c) overproduction of copper efflux pumps that secrete copper out of the cells, and (d) recruitment of copper-binding and copper-oxidizing proteins that prevent copper from interacting with cellular components (*Novoa-Aponte et al., 2019*; *Öztürk et al., 2023*; *Roy et al., 2022*; *Zuily et al., 2022*; *Dennison et al., 2018*). The genome of *Pseudomonas putida* was predicated to encode a dozen of proteins involved in copper resistance (*Cánovas et al., 2003*), including copper sensor proteins (CopS-I, CopS-II), P-type ATPases for copper efflux (CueA, CopA-I, CopA-II, CopB-I, and CopB-II), heavy-metal efflux complex components (CusA, CusB, CusC, and CusF). Expression of these proteins were regulated by copper-responsive positive regulatory proteins (CopR-I, CopR-II, CueR; *Adaikkalam and Swarup, 2002*; *Hofmann et al., 2021*). CueR regulated the expression of genes implicated in cytoplasmic copper homeostasis, whereas CopR controlled the expression of genes involved in maintaining periplasmic metal level (*Quintana et al., 2017*). Meanwhile, some proteins/systems appeared to be duplicated, and some were proved to be functionally redundant (*Adaikkalam and Swarup, 2002*; *Quintana et al., 2017*). Besides, extracellular polymeric substances (EPS) in *P. putida* biofilms showed high affinity for most heavy metals, including copper, thereby providing a protective barrier under copper stress (*Fang et al., 2011*; *Lin et al., 2020*; *Lin et al., 2018*).

In addition to the above copper resistance strategies, bacteria can avoid copper stress through chemotaxis. For example, in *Caulobacter crescentus*, the reactive oxygen species derived from cytoplasmic copper ions mediate the bacterial chemotaxis to copper, and a potential cytoplasmic MCP McpR regulates bacterial chemotaxis in response to cellular copper content, enabling bacteria to escape from copper-rich environment (*Lawarée et al., 2016*; *Louis et al., 2023*). However, unlike the widely reported mechanisms of copper resistance in diverse bacterial species, the mechanism(s) of bacterial chemotaxis to copper is poorly studied. Besides, it remains unclear how bacteria coordinate chemotaxis and resistance to copper.

Since the kinase CheA plays a central role in chemotaxis signaling, identifying CheA-interacting proteins would extend the knowledge of chemotaxis regulation. In this study, by screening proteins that interacted with CheA in *Pseudomonas putida*, we obtained 16 novel CheA-interacting proteins. Among the 16 proteins, CsoR, a copper-binding transcription regulator, inhibited the autophosphorylation of CheA, leading to decreased chemotaxis. Meanwhile, CsoR functioned as a DNA-binding repressor to inhibit the expression of copper-resistance genes. Copper-binding of CsoR relieved its inhibition of gene expression and chemotaxis.

## Results

### Identification of new CheA-interacting proteins in *P. putida*

We performed a pull-down assay to identify protein(s) interacting with CheA. Purified 6×His-tagged CheA bound onto a Ni-NTA agarose column was used as 'bait' protein to pull out potential CheA-interacting 'prey' protein(s) from whole cell lysate of *P. putida* KT2440, and a blank Ni-NTA agarose column was used as a negative control. All 'prey' proteins from the CheA-binding column and control column were collected, resolved by SDS–PAGE (*Figure 1a*), and analyzed by mass spectrometry (MS). In MS analysis, 43 proteins showed a significantly higher amount in the CheA-binding column than in the control column (Log$_2$(fold change)>2) (*Figure 1b*, *Supplementary file 1a*). As expected, the 'bait' protein CheA showed the highest amount, with a Log$_2$(fold change) value close to 8. Meanwhile, the response regulator CheY and the phosphatase CheZ (two proteins known to be associated with CheA) also showed high Log$_2$(fold change) values. In addition to these three proteins (CheA, CheY, and CheZ), the remaining 40 proteins were considered potential new CheA-interacting proteins.

To verify the above result, we performed bacterial two-hybrid (BTH) assay to test the interactions between CheA and the 40 proteins. The results from the BTH assay revealed that 19 proteins showed apparent interaction with CheA, including PP_2969 (CsoR), PP_1612 (Eno), PP_4111 (FusB), PP_5046 (GlnA), PP_1074 (GlpR), PP_4728 (GrpE), PP_0853 (IspG), PP_1877 (MsrC), PP_2378 (NfuA), PP_1023 (Pgl), PP_5006 (PhaD), PP_0691 (ProB), PP_0148, PP_1644, PP_2683, PP_3177, PP_3227, PP_3501, and PP_4460 (*Figure 1c*). Meanwhile, the rest 21 proteins displayed no significant interaction with CheA in BTH assay (*Figure 1—figure supplement 1a and b*), suggesting that they were false positive results in the pull-down assay. We further tested the interactions between CheA and the 19 proteins in *P. putida* using bimolecular fluorescence complementation (BiFC) assay. The results displayed that except three proteins (PP_2683, PP_3227, and PP_4460) showed no apparent interaction with CheA, the other 16 proteins displayed obvious interaction with CheA in the BiFC assay (*Figure 1d*, *Figure 1—figure supplement 2*). Collectively, using pull-down, BTH, and BiFC assays, we identified 16 new CheA-interacting proteins in *P. putida*, including CsoR, Eno, FusB, GlnA, GlpR, GrpE, IspG, MsrC, NfuA, Pgl, PhaD, ProB, PP_0148, PP_1644, PP_3177, and PP_3501.

### Effect of CheA-interacting proteins on bacterial chemotaxis

Since CheA played a crucial role in bacterial chemotaxis, we wondered whether the 16 CheA-interacting proteins were involved in regulating chemotaxis. Thus, we overexpressed each of the 16 proteins in wild-type *P. putida* and tested the chemotaxis ability. The wild-type *P. putida* harboring empty vector (WT +pVec) was used as the control. Chemotaxis ability was assessed using semi-solid nutrient agar plates on which bacteria formed large colonies (swimming zone) by generating and following attractant gradients leading outward from the colony origin (*Pham and Parkinson, 2011*). Growth of the 16 strains in liquid medium showed a similar trend as that of the control strain (*Figure 2—figure supplement 1*), suggesting that the 16 proteins had no noticeable effect on bacterial growth. As shown in *Figure 2a*, five strains (WT +p*csoR*, WT +p*ispG*, WT +p*nfuA*, WT +p*phaD*,

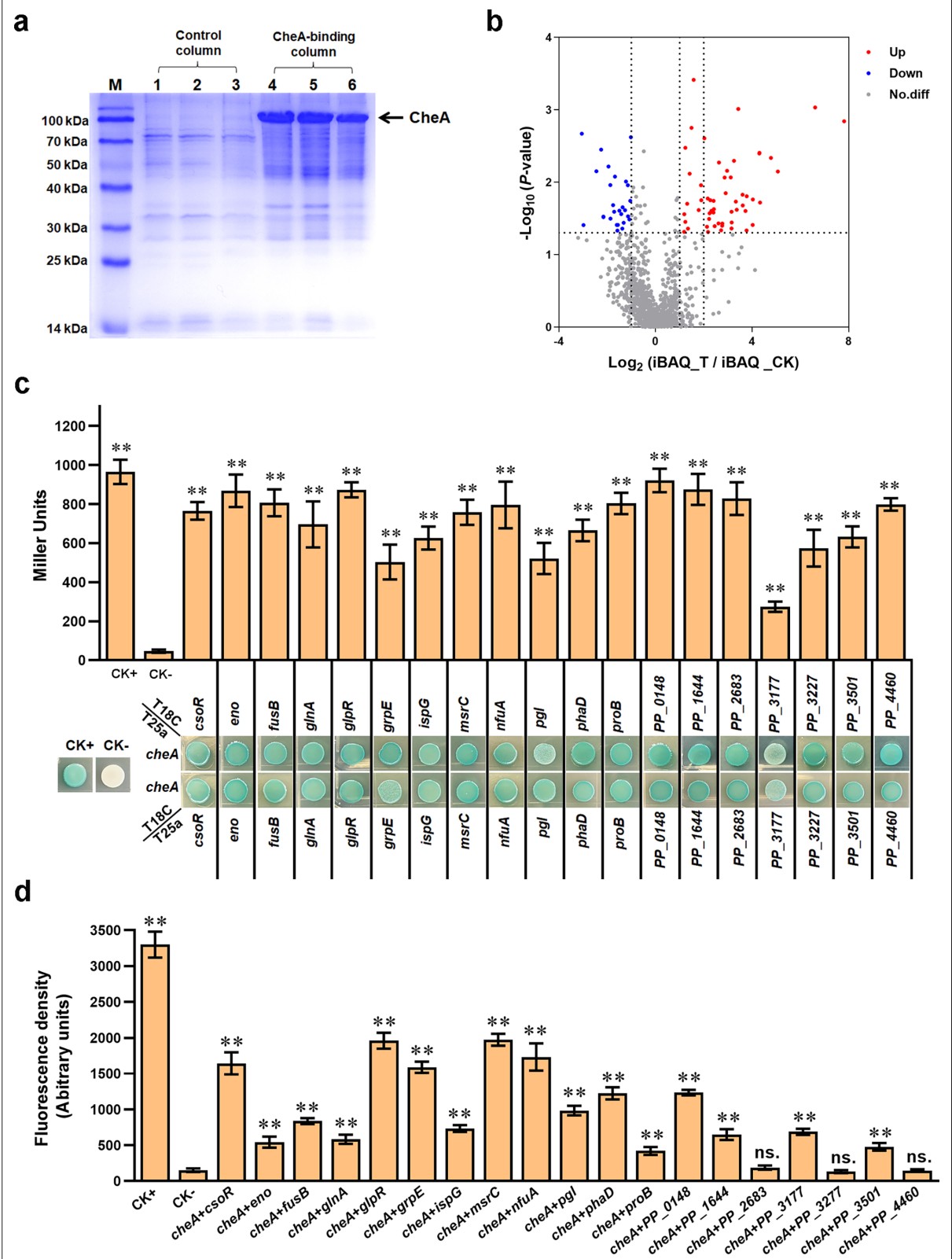

**Figure 1.** Screening and verifying proteins interacting with CheA. (**a**) Protein samples obtained in pull-down assay and detected by SDS/PAGE. The 'bait' protein CheA on the gel was indicated. Lanes 1, 2, and 3 are samples from the control column, and lanes 4, 5, and 6 are samples from the CheA-binding column. M represents a protein marker. (**b**) The volcano plot shows the p-value and fold-change of all proteins identified in MS analyses. Red spots represent proteins that showed two or higher folds in the experimental group compared with the control group (p<0.05). Blue spots

*Figure 1 continued on next page*

*Figure 1 continued*

represent proteins with a higher amount in the control group (p<0.05). Grey spot proteins showed no apparent difference between the two groups (p>0.05). (**c**) Detect the interaction between CheA and indicated proteins via BTH. Blue indicates protein-protein interaction in the colony after 60 hr of incubation, while white indicates no protein-protein interaction. A colony containing T25a-zip and T18C-zip plasmids was used as a positive control (CK+), and a colony containing empty T25a and T18C plasmids was used as a negative control (CK-). The LacZ activities of colonies were shown above the colonies. (**d**) The red fluorescence intensities in BiFC assay. The results in panels c and d are the average of three independent assays. Error bars represent standard deviations. The asterisks above the column denote significant differences (**p<0.01) between indicated strains and CK- strain. 'ns.' represents none statistically significant between indicated strain and CK- strain.

The online version of this article includes the following source data and figure supplement(s) for figure 1:

**Source data 1.** Excel file containing original SDS-PAGE gel for *Figure 1a*.

**Source data 2.** Excel file containing original SDS-PAGE gel for *Figure 1a*, indicating the relevant bands and treatments.

**Figure supplement 1.** Detect the interaction between CheA and the 40 proteins using BTH.

**Figure supplement 2.** Detect the interaction between CheA and indicated proteins using BiFC.

and WT +p*PP_1644*) displayed smaller colony than the control strain (WT +pVec), indicating a weaker chemotaxis ability in these five strains. The other 11 strains showed a similar chemotaxis ability as WT +pVec. These results suggested that five CheA-interacting proteins (CsoR, IspG, NfuA, PhaD, and PP_1644) inhibited chemotaxis in *P. putida*.

## CsoR and PhaD inhibit CheA autophosphorylation

CheA has autophosphorylation activity, and it can phosphorylate its cognate response regulator (*Muok et al., 2020*). To test whether phosphate transfer existed between CheA and the 16 proteins, we performed a phosphate transfer assay using purified proteins and $[^{32}P]ATP[\gamma P]$. We successfully purified 14 out of the 16 proteins but failed to purify two proteins (GlpR and PP_3177) after several attempts. Thus, the two proteins were not included in the phosphate transfer assay. The results showed that CheA exhibited a strong autophosphorylation signal after incubation with $[^{32}P]ATP[\gamma P]$ for 45 min (*Figure 2—figure supplement 2a*). Then, each of the 14 proteins or CheY (positive control) was added to phosphorylated CheA to investigate the phosphate transfer. The addition of CheY to phosphorylated CheA led to a labeling of CheY and a reduction in the phospholabeling of CheA (*Figure 2—figure supplement 2a*), indicating a phosphate transfer happened between CheA and CheY. However, no labeling of the 14 target proteins was observed, and there was no apparent change in the phospholabeling of CheA after adding each of the 14 proteins (*Figure 2—figure supplement 2a*), suggesting no phosphate transfer happened between CheA and the 14 tested proteins.

Then, we wondered whether the 14 proteins influenced CheA autophosphorylation. To answer this question, we mixed CheA with each of the 14 proteins before adding the substrate $[^{32}P]ATP[\gamma P]$. The mixture containing CheA and bovine serum albumin (BSA) was used as a negative control. As shown in *Figure 2b*, CsoR/PhaD significantly decreased the phospholabeling of CheA. In contrast, the other 12 proteins and BSA had no apparent influence on the phospholabeling of CheA. We further tested the impact of CsoR/PhaD on CheA autophosphorylation with a more detailed assay, in which CheA was mixed with an increased amount of CsoR/PhaD. The results showed that CheA phospholabeling decreased as CsoR/PhaD increased. In contrast, the increase of BSA had no obvious influence on CheA phospholabeling (*Figure 2c and d*). These results indicated that CsoR and PhaD inhibited CheA autophosphorylation. It was also possible that CsoR and PhaD degraded the substrate $[^{32}P]ATP[\gamma P]$ in the reaction mixture, resulting in decreased CheA autophosphorylation. To test this possibility, we examined whether CsoR and PhaD had ATPase activity. The known ATPase FleQ was used as a positive control, and BSA was used as a negative control. The results showed that adding FleQ into the reaction mixture caused a decrease in ATP level (*Figure 2—figure supplement 2*), indicating the existence of ATPase activity. Meanwhile, the addition of CsoR/PhaD exhibited no apparent influence on ATP level as the addition of BSA (*Figure 2—figure supplement 2b*), suggesting that CsoR and PhaD had no ATPase activity.

In the chemotaxis signaling pathway, CheA transfers the phosphate group to the response regulator CheY to modulate flagellar rotation (*Wadhams and Armitage, 2004*; *Porter et al., 2011*). Since CsoR and PhaD interacted with CheA, we wondered whether CsoR/PhaD influenced the phosphate transfer between CheA and CheY. Thus, we incubated CsoR/PhaD with the phosphorylated CheA

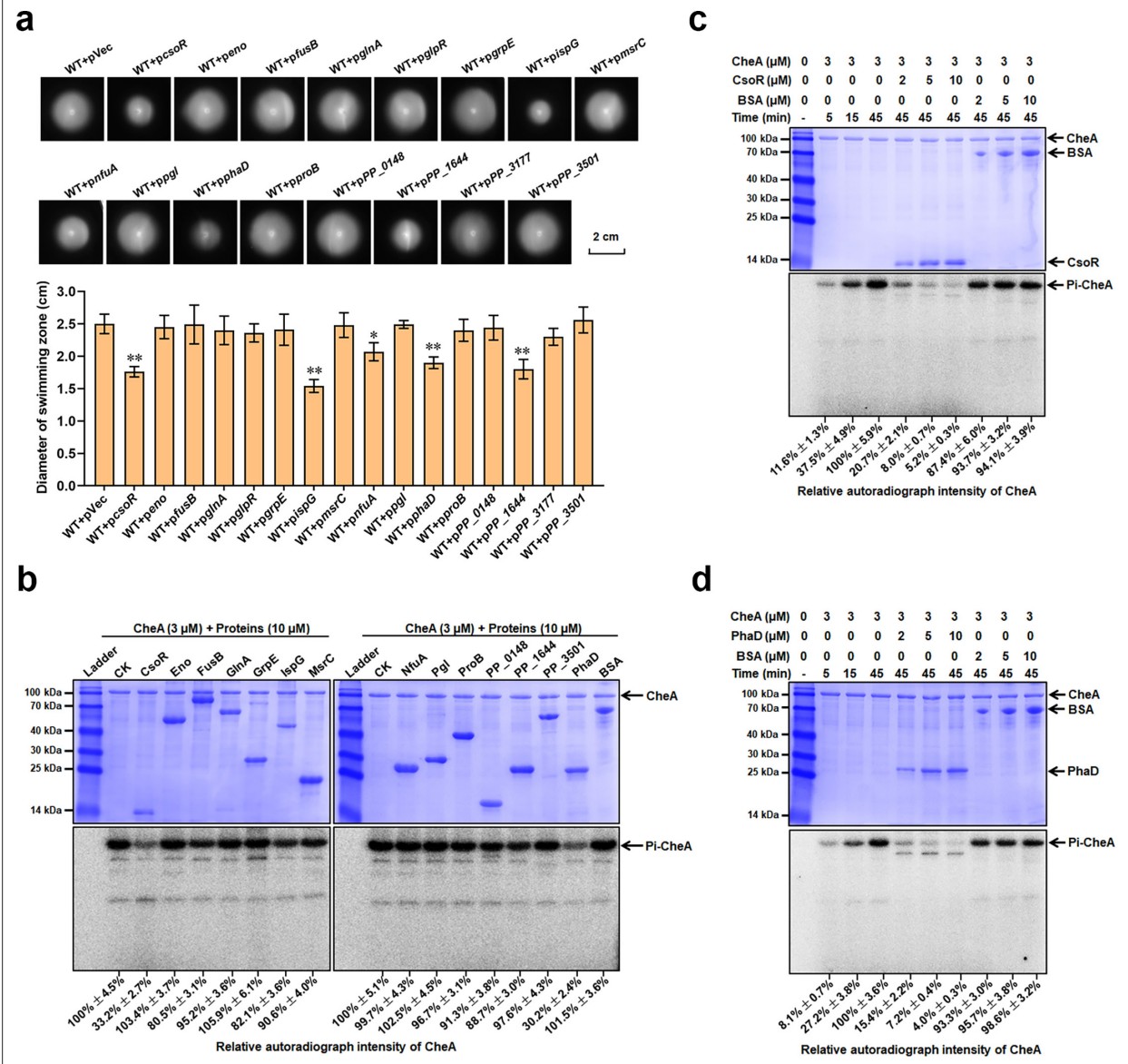

**Figure 2.** CsoR and PhaD inhibit CheA autophosphorylation. (**a**) Chemotaxis of indicated strains on semisolid plates. Photos of colonies on the top were taken after 16 hr incubation at 28 °C. Diameters of colonies (swimming zone) shown at down were calculated from three replicates. The asterisks above the column denote significant differences (*p<0.05, **p<0.01) between indicated strains and control strain (WT + pVec) analyzed by Student's t-test. (**b**) Effect of the 14 proteins on the CheA autophosphorylation. The name/ID and the concentration of tested proteins added in each lane are indicated above the gel. CK represents CheA alone in the reaction mixture. BSA is used as a negative control. The ladder represents a protein marker. (**c and d**) CsoR (**c**) and PhaD (**d**) affect CheA autophosphorylation. The concentration of tested proteins added in each lane is indicated above the gel. The time represents the time of the CheA autophosphorylation reaction. The SDS-PAGE gels in panels b, c, and d were detected by Coomassie Blue Staining (Above) and autoradiograph (Below). The experiments for panels b, c, and d were repeated three times, and a representative photo was shown. The relative autoradiograph intensity of the CheA band was calculated using Image J software and shown below each lane.

The online version of this article includes the following source data and figure supplement(s) for figure 2:

**Source data 1.** Excel file containing original SDS-PAGE gels and autoradiograph photos for *Figure 2b, c and d*.

**Source data 2.** Excel file containing original SDS-PAGE gels and autoradiograph photos for *Figure 2b, c and d*, indicating the relevant bands and treatments.

**Figure supplement 1.** Growth curve of the 16 overexpression strains and wild-type strain in liquid LB broth (100 mL in a 250 mL triangular glass flask, at 28 °C with 180 rpm shaking).

**Figure supplement 2.** Role of target proteins in the CheA-mediated transphosphorylation.

**Figure supplement 2—source data 1.** Excel file containing original SDS-PAGE gels and autoradiograph photos for *Figure 2—figure supplement 2a*

*Figure 2 continued on next page*

*Figure 2 continued*

*and c.*

**Figure supplement 2—source data 2.** Excel file containing original SDS-PAGE gels and autoradiograph photos for *Figure 2—figure supplement 2a and c*, indicating the relevant bands and treatments.

before adding CheY to the reaction mixture. The results revealed that CheA phospholabeling in the CsoR/PhaD-adding group reduced to a similar level as that in the group without CsoR/PhaD (*Figure 2—figure supplement 2c*), implying that CsoR/PhaD had no apparent influence on the phosphate transfer between CheA and CheY.

## The domains of CheA involved in interacting with CsoR and PhaD

To further explore how CsoR and PhaD affect CheA autophosphorylation, we determined the domain(s) of CheA involved in interacting with CsoR and PhaD. Similar to the *E. coli* CheA, the *P. putida* CheA consists of five domains with distinct functions (*Figure 3a*). We constructed five truncated CheA variants, with each missing one domain (termed CheA$_{\Delta HPT}$/CheA$_{\Delta YB}$/CheA$_{\Delta Dim}$/CheA$_{\Delta HATPase}$/CheA$_{\Delta WB}$; *Figure 3a*). Then we performed BTH assay to test the interaction between CsoR/PhaD and each of these truncated CheAs. The results revealed that three truncated CheAs (CheA$_{\Delta HPT}$, CheA$_{\Delta Dim}$, and CheA$_{\Delta HATPase}$) showed no interaction with CsoR and PhaD. In comparison, the other two truncated CheAs (CheA$_{\Delta YB}$ and CheA$_{\Delta WB}$) interacted with CsoR and PhaD with a similar intensity as the wild-type CheA did (*Figure 3b and c*), indicating that the HPT (P1), Dimer (P3), and HATPase_c (P4) domains were essential for interacting with CsoR and PhaD, while the YB (P2) and WB (P5) domains were not required to interact with CsoR and PhaD. To further test this result, we cloned each of the five domains

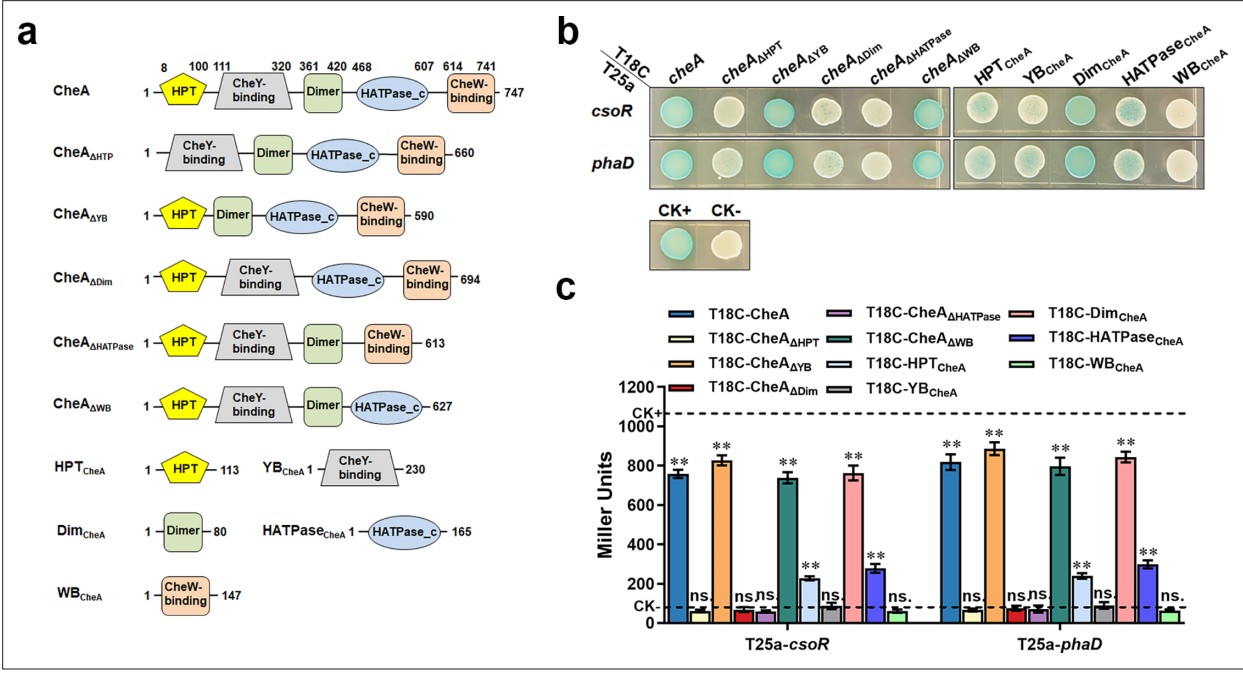

**Figure 3.** CheA domains involved in interacting with CsoR and PhaD. (**a**) Schematic diagram of CheA and truncated CheA proteins. The predicted domains are based on the Pfam database and the amino acid positions where the predicted domains start and end are shown. (**b**) The interaction between CheA domains and CsoR/PhaD was tested using BTH. Blue indicates protein-protein interaction in the colony after 60 h of incubation, while white indicates no protein-protein interaction. A colony containing T25a-zip and T18C-zip plasmids was used as a positive control (CK+), and a colony containing empty T25a and T18C plasmids was used as a negative control (CK-). (**c**) Confirmation of BTH interactions in panel B by LacZ activity assay. The results are the average of three independent assays. Error bars represent standard deviations. The asterisks above the column denote significant differences (**p<0.01) between indicated strains and CK- strain analyzed by Student's t-test. 'ns.' represents none statistically significant between indicated strain and CK- strain.

The online version of this article includes the following figure supplement(s) for figure 3:

**Figure supplement 1.** Predicted structure of the *P. putida* CsoR.

into a BTH vector and tested the interaction between CsoR/PhaD and each of the five domains. The result showed that the Dimer domain (Dim$_{CheA}$) interacted with CsoR and PhaD like the whole-length CheA did (*Figure 3b and c*). The HPT domain (HPT$_{CheA}$) and the HATPase_c domain (HATPase$_{CheA}$) showed a weaker interaction with CsoR and PhaD. Meanwhile, the YB domain (YB$_{CheA}$) and the WB domain (WB$_{CheA}$) displayed no interaction with CsoR and PhaD (*Figure 3b and c*). Together, these results revealed that the Dimer domain (P3) of CheA played a significant role, the HPT (P1) and HATPase_c (P4) domains played a minor role, while the CheY-binding (P2) and CheW-binding (P5) domains played no role in the interaction between CheA and CsoR/PhaD. Since the three domains involved in interacting with CsoR/PhaD were also essential for CheA autophosphorylation activity (*Bilwes et al., 1999*; *Muok et al., 2020*), CsoR/PhaD might inhibit CheA autophosphorylation by inhibiting the function of the three domains.

Of the two proteins that inhibit CheA autophosphorylation, CsoR is annotated as a metal-binding transcriptional repressor, and PhaD is annotated as a TetR family transcriptional regulator (*Winsor et al., 2016*.) Although the two proteins were quite different in size (CsoR 10.8 kDa, PhaD 23.1 kDa), they interacted with the same domains of CheA to inhibit its autophosphorylation, we wondered whether the two proteins shared similarity in sequence and structure. BLAST result revealed no sequence similarity between CsoR and PhaD (data not shown). We predicted structures of CsoR and PhaD using AlphaFold (*Figure 3—figure supplement 1a and b*), and compared their structures using Pymol. However, the result revealed no significant structural similarity between the two proteins (*Figure 3—figure supplement 1c*).

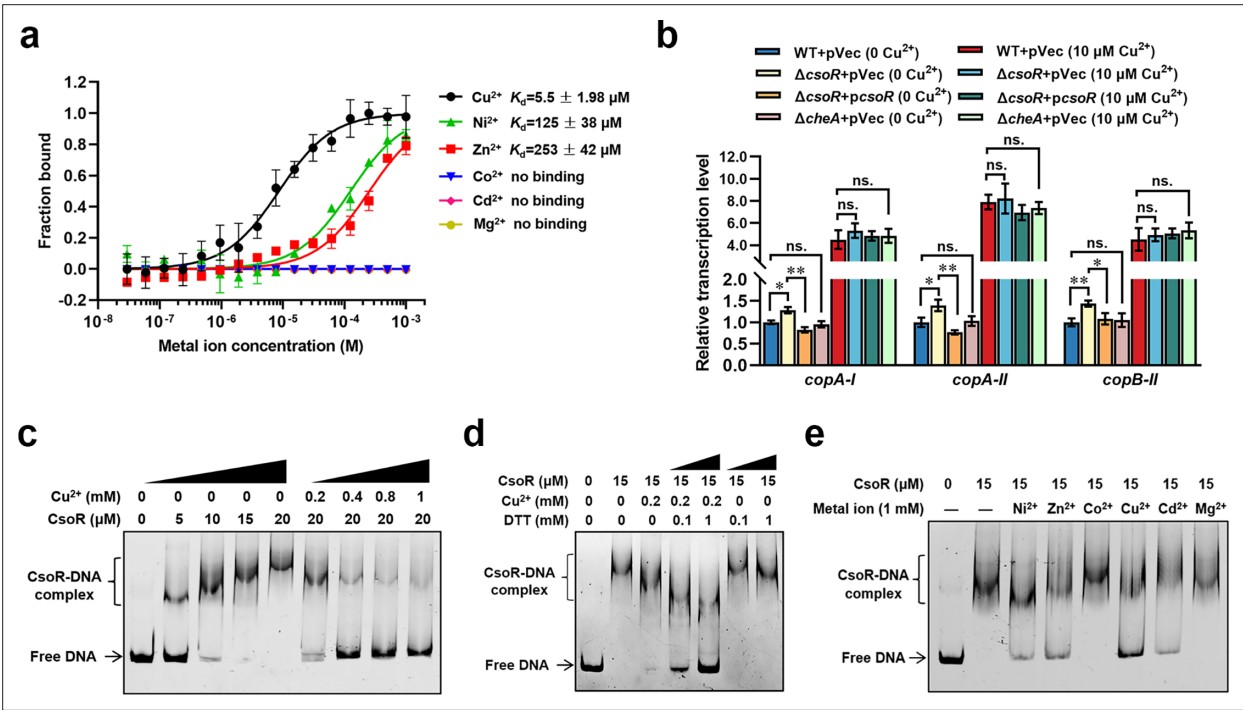

**Figure 4.** CsoR is a metal-binding repressor. (**a**) MST analysis of the interaction between CsoR-GFP and metal ions. CsoR-GFP (250 nM) was incubated with increasing concentrations of metal ions. (**b**) Analysis of relative transcription level of target genes in wild-type (WT +pVec), *csoR* mutant (Δ*csoR* +pVec), complemented strain (Δ*csoR* +p*csoR*), and *cheA* mutant (Δ*cheA* +pVec) in the presence and absence of CuCl₂ (10 µM) by qRT-PCR. The results are the average of three independent assays. Error bars represent standard deviations. The asterisks represent statistically significant differences between the two compared strains (*p<0.05, **p<0.01). 'ns.' represents none statistically significant between two compared strains. (**c**) Analysis for interactions between CsoR and *copA-I* promoter DNA using EMSA. (**d**) The effect of DTT/CuCl₂ +DTT on the interaction between CsoR and *copA-I* promoter DNA. (**e**) The effect of indicated metal ions on the interaction between CsoR and *copA-I* promoter DNA. The concentrations of CsoR, metal ions, and DTT in panels c, d, and e added in each lane are shown above the gel. Free DNA and CsoR-DNA complex are indicated.

The online version of this article includes the following source data and figure supplement(s) for figure 4:

**Source data 1.** Excel file containing original EMSA Native-PAGE gels for *Figure 4c, d and e*.

**Source data 2.** Excel file containing original EMSA Native-PAGE gels for *Figure 4c, d and e*, indicating the relevant bands and treatments.

**Figure supplement 1.** Function of CsoR in the expression of metal resistant genes and the bacterial growth under copper stress.

# CsoR is a metal-binding transcriptional repressor for metal-resistance genes

Previous research revealed that the expression of CsoR (also named MreA) was induced by cadmium, nickel, zinc, and cobalt (*Haritha et al., 2009*). The homolog of CsoR in *Mycobacterium tuberculosis* binds to copper and regulates the expression of copper-resistance genes (*Liu et al., 2007*; *Marcus et al., 2016*). However, the metal-binding ability and the function of the *P. putida* CsoR were not experimentally characterized. Thus, we tested the metal-binding ability of the *P. putida* CsoR using MicroScale Thermophoresis (MST). Six metal ions were involved in the assay, including copper ($Cu^{2+}$), zinc ($Zn^{2+}$), nickel ($Ni^{2+}$), cobalt ($Co^{2+}$), cadmium ($Cd^{2+}$), and magnesium ($Mg^{2+}$). The results revealed that CsoR bound to three out of the six tested metal ions ($Cu^{2+}$, $Zn^{2+}$, $Ni^{2+}$), and the binding to $Cu^{2+}$ was the strongest with a calculated binding constant ($K_d$) of 5.5±1.98 µM. In contrast, the binding to $Ni^{2+}$/$Zn^{2+}$ was weak (with $K_d$ value of 125±38 µM and 253±42 µM, respectively) (*Figure 4a*). Meanwhile, CsoR showed no apparent binding to $Co^{2+}$/$Cd^{2+}$/$Mg^{2+}$ under the same condition (*Figure 4a*). We further constructed a *csoR* deletion mutant (Δ*csoR*) and investigated the effect of *csoR* deletion on the expression of copper resistance genes using quantitative real-time PCR (qRT-PCR). Three key copper resistance genes (*copA-I*, *copA-II*, and *copB-II*) from two operons were chosen as targets in qRT-PCR. The results showed that *csoR* deletion (Δ*csoR* +pVec) led to a weak but significant increase (about 1.3-fold) in the expression of the three genes, and complementation (Δ*csoR* +p*csoR*) decreased the expression of the three genes (*Figure 4b*). The addition of CuCl₂ (final concentration 10 µM) induced the expression of the three genes (about fivefold; *Figure 4b*). However, no noticeable difference in gene expression level was observed between WT +pVec and Δ*csoR* +pVec, as well as between Δ*csoR* +pVec and Δ*csoR* +p*csoR* (*Figure 4b*), implying that CsoR was not required for the copper-induced genes expression. Besides, the deletion of *cheA* (Δ*cheA* +pVec) displayed no obvious effect on the expression of the three genes in either the presence or absence of copper (*Figure 4b*). Since CsoR also bound $Zn^{2+}$ and $Ni^{2+}$, we investigated the influence of CsoR on the expression of several other metal resistance genes, including three nickel-resistance genes (*nikA*, *nikB*, and *nikD*), one zinc-resistance genes (*znuC*), two cadmium-resistance genes (*cadA-I* and *cadA-III*), two cobalt-resistance gene (*cbiD* and *cbtA*), and three multiple metal-resistance genes (*czcC-I*, *czcB-II*, and *PP_0026*). The results reveled that CsoR repressed the expression of *nikB*, *cadA-I*, *cadA-III*, *cbtA*, *czcC-I*, *czcB-II*, and *PP_0026*, and the inhibition degree was close to the inhibition degree of copper resistance genes (*Figure 4—figure supplement 1a*). Together, these results demonstrated that CsoR functioned as a metal-binding repressor for metal resistance genes in *P. putida*.

Previous studies reported that CsoR bound the promoter of target genes to inhibit gene expression (*Liu et al., 2007*; *Marcus et al., 2016*). We further tested the interaction between CsoR and the promoter of *copA-I* using electrophoretic mobility shift assay (EMSA). The fragment of *copA-I* promoter exhibited a stepwise increase in the shifted DNA amount (CsoR-DNA complex), with the CsoR protein amount increasing from 5 to 20 µM (*Figure 4c*). Adding $Cu^{2+}$ to the reaction mixture decreased the CsoR-DNA complex (*Figure 4c*), indicating that $Cu^{2+}$ inhibited the interaction between CsoR and promoter DNA. Previous study in *M. tuberculosis* showed that CsoR bound a single-monomer mole equivalent of $Cu^+$ to form a trigonally coordinated complex (*Liu et al., 2007*), but our results indicated that CsoR bound to $Cu^{2+}$. To further test whether $Cu^+$ bind to CsoR and affect its DNA-binding ability, we added dithiothreitol (DTT) to the EMSA reaction mixture. DTT can reduce $Cu^{2+}$ to $Cu^+$ in solution (*Kręźel et al., 2001*). As shown in the following *Figure 4d*, the addition of DTT (0.1 and 1 mM) decreased CsoR-DNA interaction in the presence of 0.2 mM $Cu^{2+}$, and the addition of DTT alone had no apparent influence on CsoR-DNA interaction, indicating that DTT enhanced the inhibition of $Cu^{2+}$ on CsoR-DNA interaction. These results suggested that the $Cu^+$ converted from $Cu^{2+}$ by DTT had stronger inhibitory effect than $Cu^{2+}$ on CsoR-DNA interaction, indicating that CsoR bound more strongly to $Cu^+$ than to $Cu^{2+}$. Besides, $Ni^{2+}$, $Zn^{2+}$, and $Cd^{2+}$ also exhibited an inhibitory effect on the interaction between CsoR and promoter DNA, but to a much lower extent compared with $Cu^{2+}$ (*Figure 4e*). Meanwhile, $Co^{2+}$ and $Mg^{2+}$ displayed no obvious effect on the CsoR-DNA interaction (*Figure 4e*). These results showed that CsoR was a promoter-binding transcriptional repressor, and binding to metal (especially copper) decreased the interaction between CsoR and promoter DNA.

To test the effect of CsoR and CheA on bacterial copper resistance, we tested the growth of Δ*csoR* and Δ*cheA* under different copper concentrations using both solid agar plate and liquid medium. However, in both cases, there was no significant difference between the growth trend of Δ*csoR*/Δ*cheA*

and the WT strain at different copper concentrations (*Figure 4—figure supplement 1b and c*). This might attribute to the fact that CsoR was a repressor, and the expression of copper resistance genes in WT was similar to that in Δ*csoR* under copper stress (*Figure 4b*).

## Copper inhibits the interaction between CheA and CsoR

Since CsoR interacted with CheA and bound copper, we wondered whether copper affected the interaction between CheA and CsoR. Thus, we investigated the interaction between CheA and CsoR under different $CuCl_2$ concentrations using MST. As revealed in *Figure 5a*, CheA showed strong interaction with CsoR in the absence of $Cu^{2+}$ ($K_d = 0.17 \pm 0.1$ μM), and the addition of $Cu^{2+}$ (20 and 200 μM) led to increased binding constant ($0.59 \pm 0.2$ μM and $2.15 \pm 0.97$ μM), indicating that copper decreased the interaction between CheA and CsoR. A similar trend was observed in the pull-down assay, in which the amount of CsoR bound by CheA gradually decreased with the concentration of $Cu^{2+}$ increased from 2 to 20 μM (*Figure 5b*).

A previous study on *M. tuberculosis* CsoR revealed three residues that played a vital role in copper-binding (*Liu et al., 2007*). The alignment assay showed that the three residues were conserved among CsoR homologs from several bacterial species, including the *P. putida* CsoR (*Figure 5—figure supplement 1a*). To test the role of the three residues in copper-binding, we individually replaced each of the three residues (Cys40, His65, and Cys69) with alanine and tested the copper-binding ability of these mutated CsoRs ($CsoR_{C40A}$, $CsoR_{H65A}$, and $CsoR_{C69A}$). MST results showed that the $CsoR_{C69A}$ displayed significantly decreased copper-binding ability compared to wild-type CsoR. In contrast, $CsoR_{C40A}$ and $CsoR_{H65A}$ showed a slight decrease in copper-binding ability compared with that of the wild-type CsoR (*Figure 5—figure supplement 1b*), suggesting that the Cys69 residue of CsoR played a critical positive role in copper-binding. Besides, the results from EMSA also supported this conclusion, in which the mutation of C69A, but not C40A/H65A, noticeably inhibited the effect of $Cu^{2+}$ on CsoR-DNA interaction (*Figure 5—figure supplement 1c and d*). BTH assay revealed that the three point-mutated CsoRs interacted with CheA with a similar intensity to the wild-type CsoR (*Figure 5—figure supplement 1e*). Results from the pull-down assay showed that the addition of $Cu^{2+}$ significantly decreased the amount of $CsoR_{C40A}$ and $CsoR_{H65A}$ bound by CheA, but had less effect on the amount of $CsoR_{C69A}$ bound by CheA under the same condition (*Figure 5b*). Together, these results revealed that copper bound CsoR and inhibited its interaction with CheA.

## Copper relives the inhibition of CsoR on bacterial chemotaxis

Since CsoR interacted with CheA and inhibited its autophosphorylation, and copper hindered the interaction between CheA and CsoR, we speculated that copper might relieve the inhibition of CsoR on CheA autophosphorylation. To test this hypothesis, we added CsoR and copper to the reaction mixture containing CheA and then analyzed the autophosphorylation of CheA using [$^{32}$P]ATP[γP]. CsoR alone significantly inhibited CheA autophosphorylation, while adding CsoR and copper showed weaker inhibition on CheA autophosphorylation (*Figure 5c*). Besides, $CsoR_{C69A}$ showed similar inhibitory effects on CheA autophosphorylation in the presence and absence of $Cu^{2+}$ (*Figure 5c*), indicating that binding with CsoR was the prerequisite for $Cu^{2+}$ to relieve the inhibition of CsoR on CheA autophosphorylation.

We further tested the effect of CsoR on chemotaxis in the presence of copper using the semisolid plate. As shown in *Figure 5d*, in the absence of $Cu^{2+}$, the overexpression of point-mutated CsoR ($CsoR_{C40A}/CsoR_{H65A}/CsoR_{C69A}$) led to a similar decrease (about 40%) in chemotaxis as the overexpression of wild-type CsoR. In comparison, in the presence of 200 μM $Cu^{2+}$, the inhibitory effect of CsoR/$CsoR_{C40A}/CsoR_{H65A}$ overexpression on chemotaxis was weaker (about 20%). However, the inhibitory effect of $CsoR_{C69A}$ overexpression on chemotaxis was not affected by $Cu^{2+}$ (*Figure 5d*). These results demonstrated that binding copper relieved the inhibition of CsoR on bacterial chemotaxis.

## CsoR inhibits chemorepellent response to copper

Excess copper is toxic to cells, and bacteria avoid high copper concentrations through chemotaxis. Since CsoR interacted with CheA and inhibited chemotaxis, we wondered about the role of CsoR in bacterial chemotaxis to copper. Thus, we tested the chemotaxis response of *P. putida* to copper gradient and the role of CsoR and CheA in this response using semisolid plates. The copper gradient was achieved by placing an agar plug containing 200 mM $CuCl_2$ in the center of a semisolid plate,

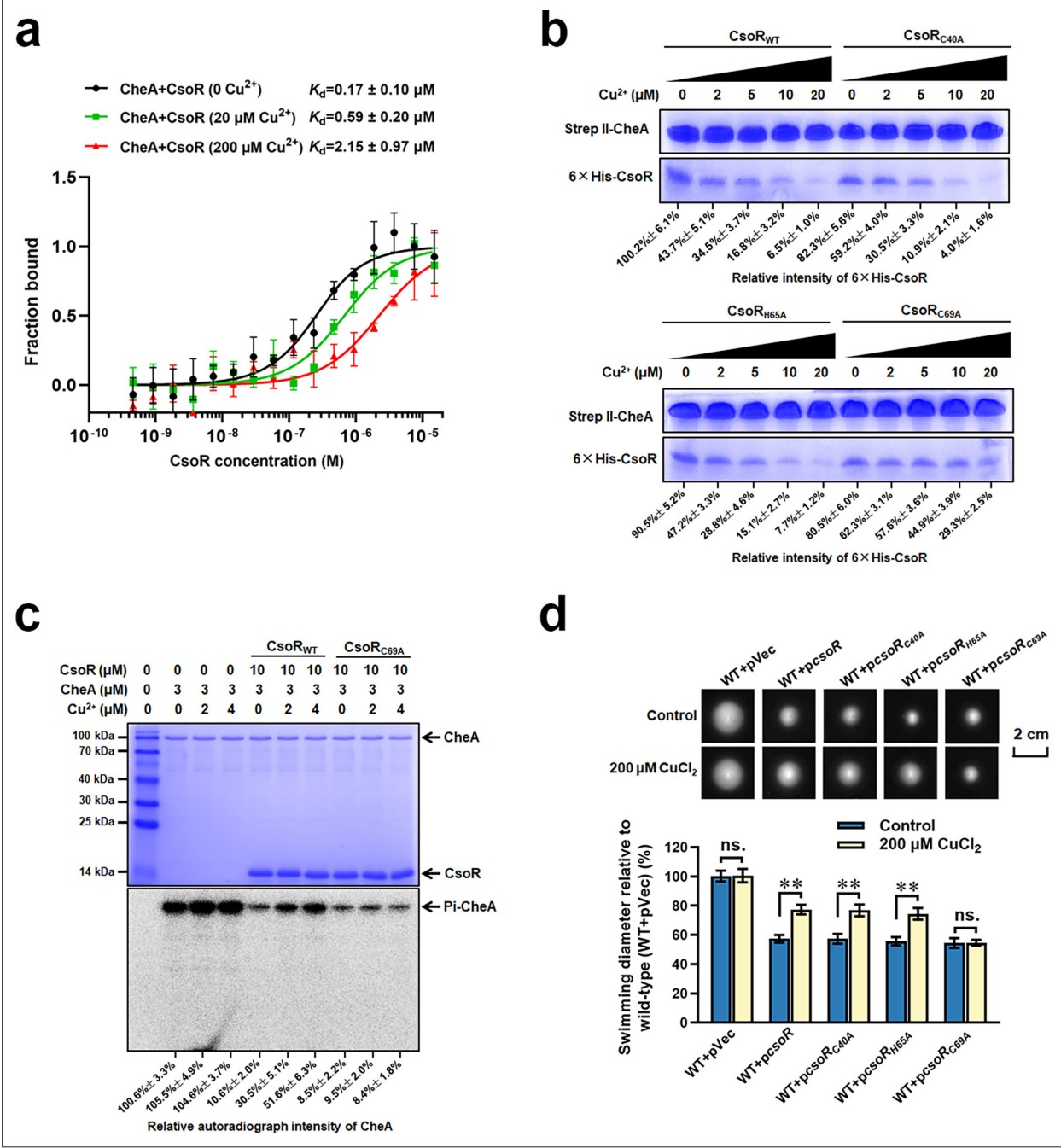

**Figure 5.** Copper inhibits the interaction between CheA and CsoR. (**a**) MST analysis of the interaction between CheA-GFP and CsoR in the presence of $Cu^{2+}$. CheA-GFP (160 nM) was incubated with increasing concentrations of CsoR. (**b**) SDS-PAGE detected protein samples obtained in a pull-down assay. The 'bait' protein Strep-CheA and the 'prey' protein His-CsoR on the gel were indicated. The gel showed the influence of $Cu^{2+}$ on the amount of 'prey' protein His-CsoR. The concentration of $Cu^{2+}$ added in each pull-down assay was displayed above the gel. The SDS-PAGE gel was detected by Coomassie Blue Staining. (**c**) CheA autophosphorylation in the presence of CsoR and $Cu^{2+}$. The tested proteins and $Cu^{2+}$ concentrations added in each lane are indicated above the gel. The SDS-PAGE gel was detected by Coomassie Blue Staining (Above) and autoradiograph (Below). The relative intensity of the CsoR band in panel b and the autoradiograph intensity of the CheA band in panel c were calculated using Image J software and shown below each lane. (**d**) Chemotaxis of indicated strains on semisolid plates supplied with or without $CuCl_2$. Photos of colonies on the top were taken after 16 hr (for the control plate) or 18 hr (for the copper-adding plate) incubation at 28 °C. The diameters of colonies were measured and normalized to the diameters of WT + pVec, shown below. The asterisks above the column denote significant differences (**$p<0.01$) between two indicated strains analyzed by Student's t-test. 'ns.' represents none statistically significant between two compared strains.

The online version of this article includes the following source data and figure supplement(s) for figure 5:

*Figure 5 continued on next page*

*Figure 5 continued*

**Source data 1.** Excel file containing original SDS-PAGE gels and autoradiograph photo for *Figure 5b and c*.

**Source data 2.** Excel file containing original SDS-PAGE gels and autoradiograph photo for *Figure 5b and c*, indicating the relevant bands and treatments.

**Figure supplement 1.** Role of the three conserved residues in the Cu²⁺-binding ability of CsoR.

**Figure supplement 1—source data 1.** Excel file containing original EMSA Native-PAGE gels for *Figure 5—figure supplement 1c and d*.

**Figure supplement 1—source data 2.** Excel file containing original EMSA Native-PAGE gels for *Figure 5—figure supplement 1c and d*, indicating the relevant bands and treatments.

and bacterial cells were spotted two centimeters away from the plug to test their chemotaxis to Cu²⁺. An agar plug without CuCl₂ was also placed in the center of a semisolid plate and used as a negative control. As shown in *Figure 6a*, the swimming zone of all tested strains was a circle shape in the control plate without CuCl₂. However, all swimming zones showed an oval shape in the plate with CuCl₂ gradient (*Figure 6a*), in which the bacterial movement distance near the plug (D1) was short, and the movement distance from the plug (D2) was long, indicating that the strains showed chemorepellent response to CuCl₂. The RI value (Response index value, RI = D1/(D1 +D2)) was further calculated to characterize the strength of the chemorepellent response to CuCl₂. The results showed that WT +p*csoR* displayed a higher RI value (0.428±0.015) than WT + pVec (0.373±0.021; *Figure 6b*). Meanwhile, Δ*csoR* + pVec showed a lower RI value (0.324±0.013) than WT + pVec, and complementation increased the RI value to wild-type level (*Figure 6b*). These results suggested that CsoR inhibited the chemorepellent response to copper. Besides, both *cheA* deletion mutant (Δ*cheA* +pVec) and *cheA csoR* double deletion mutant (Δ*csoR*Δ*cheA* + pVec) displayed no chemotaxis ability in either the presence or absence of copper gradient (*Figure 6a and b*), indicating that CsoR inhibited chemotaxis ability in a CheA-dependent manner.

Using time-lapse microscopy experiments and cell-tracking analysis, we further examined the bacterial chemorepellent response to Cu²⁺. In the control group without chemokine (Cu²⁺ gradient), cells of all tested strains swam randomly in all directions (*Figure 6c*). The center of mass (defined as the average of all single cell endpoints, and it reflects the movements of target strain and the strength of chemotaxis response) of all tested strains showed no apparent difference (*Figure 6d*). In the group with chemokine (Cu²⁺ gradient), cells of WT + pVec, WT + p*csoR*, Δ*csoR* + pVec, and Δ*csoR* + p*csoR* migrated towards the lower concentration of Cu²⁺ (*Figure 6c*). In comparison, cells of Δ*cheA* + pVec and Δ*csoR*Δ*cheA* + pVec still swam randomly in all directions (*Figure 6c*), indicating that *P. putida* cells showed a chemorepellent response to Cu²⁺ in a CheA-dependent manner. The center of mass value of WT +p*csoR* (13.85±1.92 μm) was smaller than that of WT + pVec (21.77±3.60 μm; *Figure 6d*). Meanwhile, the center of mass value of Δ*csoR* + p*csoR* (14.16±1.41 μm) was smaller than that of Δ*csoR* +pVec (21.39±2.02 μm; *Figure 6d*), suggesting that CsoR inhibited bacterial chemorepellent response to Cu²⁺. The velocities (cell migration speeds) of cells from WT +p*csoR*, Δ*csoR* + pVec, and Δ*csoR* + p*csoR* were similar to that from WT + pVec both in the presence and absence of Cu²⁺ gradient, implying that CsoR had no evident influence on bacterial migration speed. In contrast, the velocities of cells from Δ*cheA* + pVec and Δ*csoR*Δ*cheA* + pVec were lower than that from WT +pVec (*Figure 6e*), indicating that CheA played a positive role in bacterial migration speed. Together, these results demonstrated that CsoR inhibited the chemorepellent response to copper in a CheA-dependent manner.

## The interaction between CheA and CsoR exists in several bacterial species

The role of CsoR in regulating copper resistance has been reported in several bacterial species, including *Acidithiobacillus caldus* (*Hou et al., 2021*), *Bacillus subtilis* (*Smaldone and Helmann, 2007*), *Bradyrhizobium diazoefficiens* (*Pacheco et al., 2023*), *Corynebacterium glutamicum* *Teramoto et al., 2015*, *Listeria monocytogenes* (*Corbett et al., 2011*), *M. tuberculosis* (*Marcus et al., 2016*), *Staphylococcus aureus* (*Baker et al., 2011*), *Streptomyces lividans* (*Dwarakanath et al., 2012*), and *Thermus thermophilus* (*Sakamoto et al., 2010*). BLAST results showed that four of the nine above species (*A. caldus*, *B. diazoefficiens*, *B. subtilis*, and *L. monocytogenes*) had both *cheA* and *csoR* on their genomes. Besides, in addition to *P. putida*, *cheA* and *csoR* coexist in other *Pseudomonas* species, including

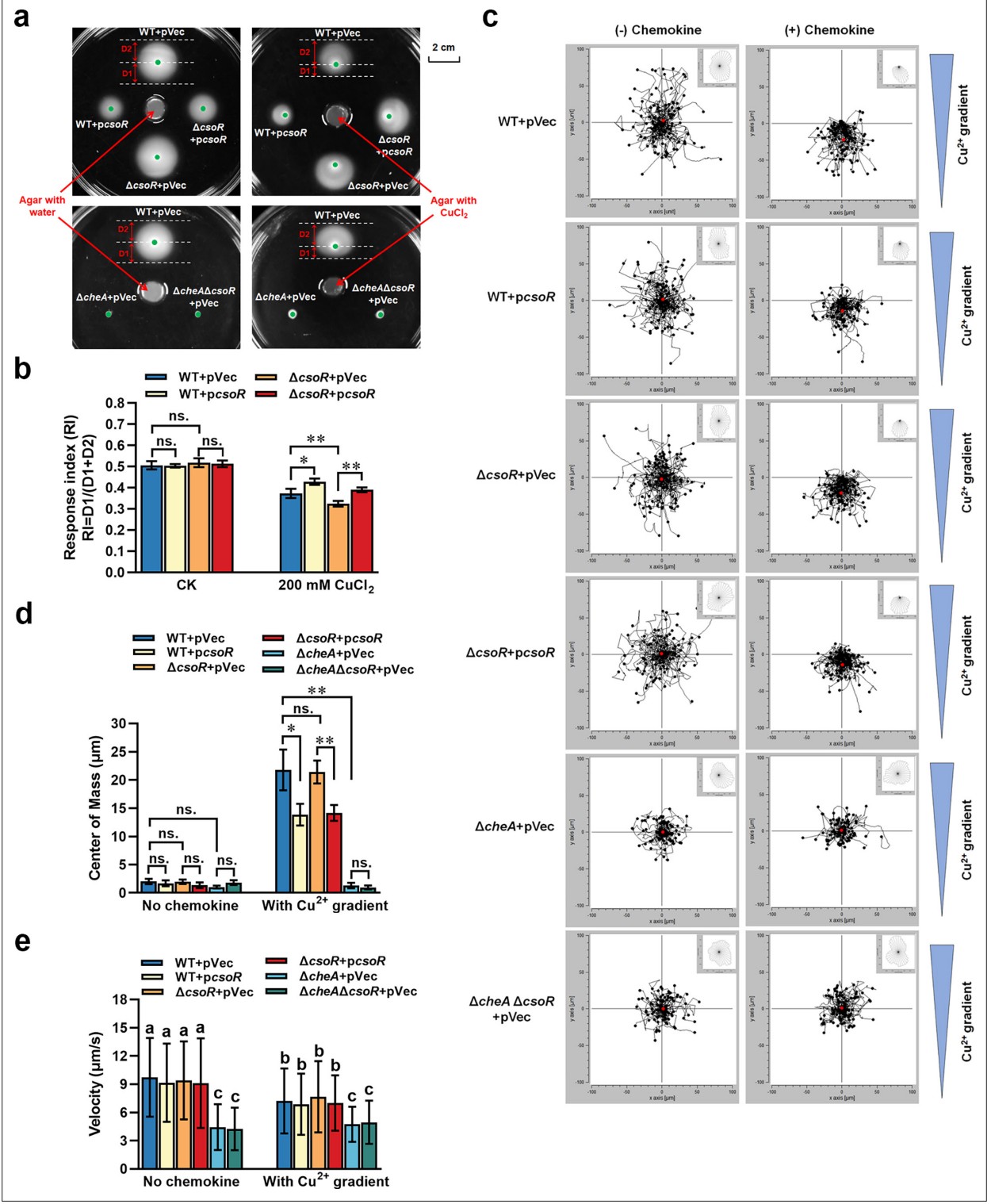

**Figure 6.** Role of CsoR in bacterial chemotaxis to copper. (**a**) Chemotaxis of indicated strains in the absence and presence of copper gradient. The chemotaxis rings and chemotaxis distance (D1/D2) were indicated by arrows. The red arrows pointed at the agar plug with or without copper in the center of the plate. The green dots represented the sites where the bacteria were initially inoculated on the semisolid plate. The assay was performed with three repeats, and a representative photo was shown. (**b**) RI value (D1/(D1 + D2)) of indicated strains shown in panel a. (**c**) Aggregated trajectories of individual tested strain cells in the absence and presence of copper gradient. The tracking data presented is a composite of two experiments performed in duplicate (n=100 cells). The overall directionality of migration is depicted in the rose diagram in the upper right corner of each single-track summary. (**d**) Center of mass of tested strains in the presence of copper gradients. It represents the average of all single-cell endpoints. The results of

*Figure 6 continued on next page*

*Figure 6 continued*

panels b and d are the average of three independent assays. Error bars represent standard deviations. The asterisks represent statistically significant differences between the two indicated strains (*$p<0.05$, **$p<0.01$). 'ns.' represents no statistically significant between the two indicated strains. (**e**) Velocity analysis of indicated strains in the presence or absence of copper gradient (n=100 cells). The lowercase letters above each bar in panel e indicate significant differences ($p<0.05$).

*Pseudomonas fluorescens*, *Pseudomonas syringae*, and *Pseudomonas stutzeri*. We wondered whether the CheA-CsoR interaction also occurred between proteins from these strains. Thus, we tested the interaction between CheA and CsoR of the same strain via BTH assay. The result showed that the CheA-CsoR interaction existed between proteins from *A. caldus*, *B. subtilis*, *P. syringae*, and *P. stutzeri* (**Figure 7**). However, CheA and CsoR from *B. diazoefficiens*, *L. monocytogenes*, and *P. fluorescens* showed no apparent interaction (**Figure 7**). Besides, the intensity of CheA-CsoR interaction was more vigorous between the two proteins from *B. subtilis*, but weaker between that from *A. caldus*, *P. syringae*, and *P. stutzeri* (**Figure 7**). These results suggested that except in *P. putida*, the interaction between CheA and CsoR also existed in other bacterial species.

## Discussion

Integrating components from different systems provides a straightforward mechanism for coordinating signaling from various systems. This study identified an interaction between the chemotaxis kinase CheA and the copper-responsive transcriptional repressor CsoR in *P. putida*. Further analysis

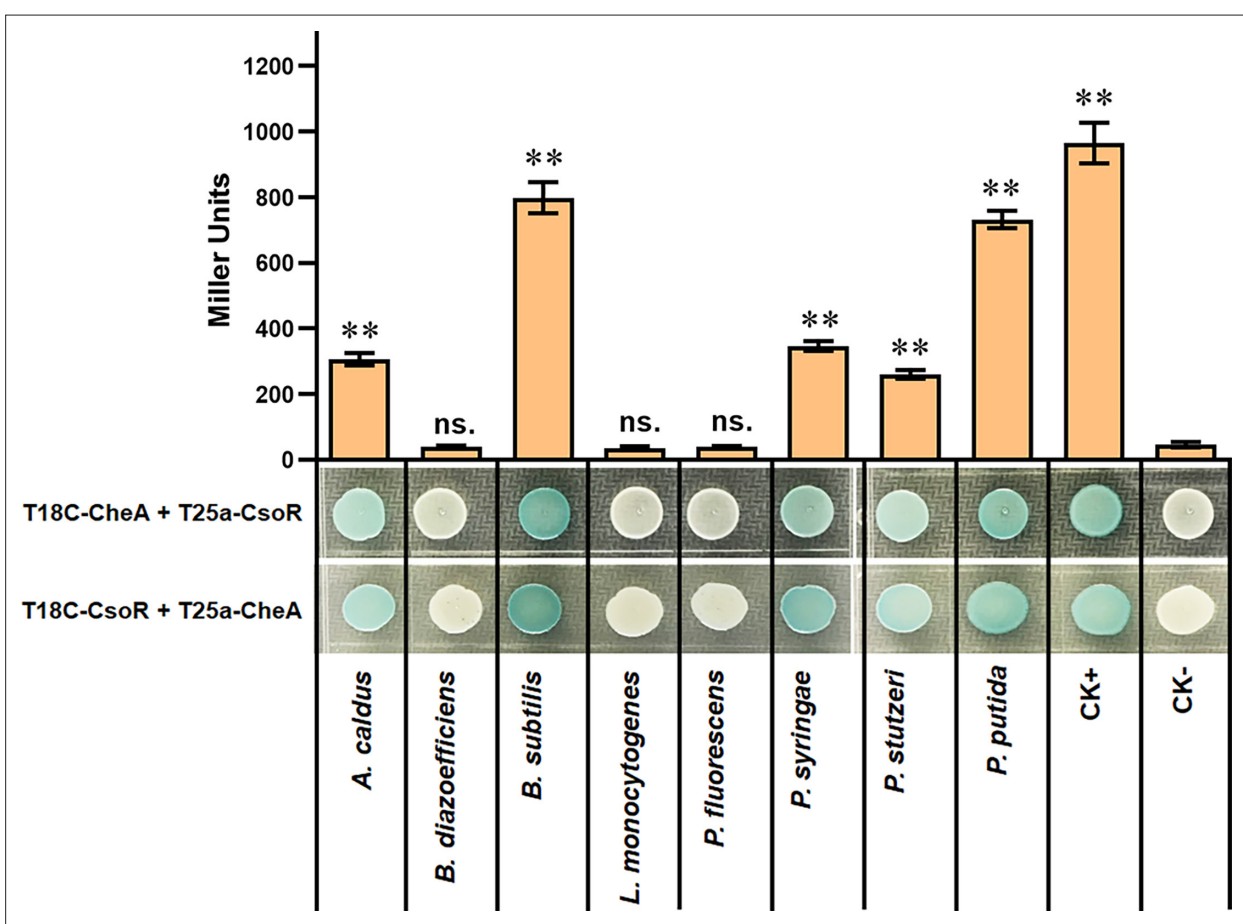

**Figure 7.** The interaction between CheA and CsoR from indicated bacterial species. The interaction between CsoR and CheA was tested by using BTH. The LacZ activities of colonies were shown above the colonies. The results are the average of three independent assays. Error bars represent standard deviations. The asterisks above the column denote significant differences (**$p<0.01$) between indicated strains and CK- strain analyzed by Student's t-test. 'ns.' represents none statistically significant between indicated strain and CK- strain.

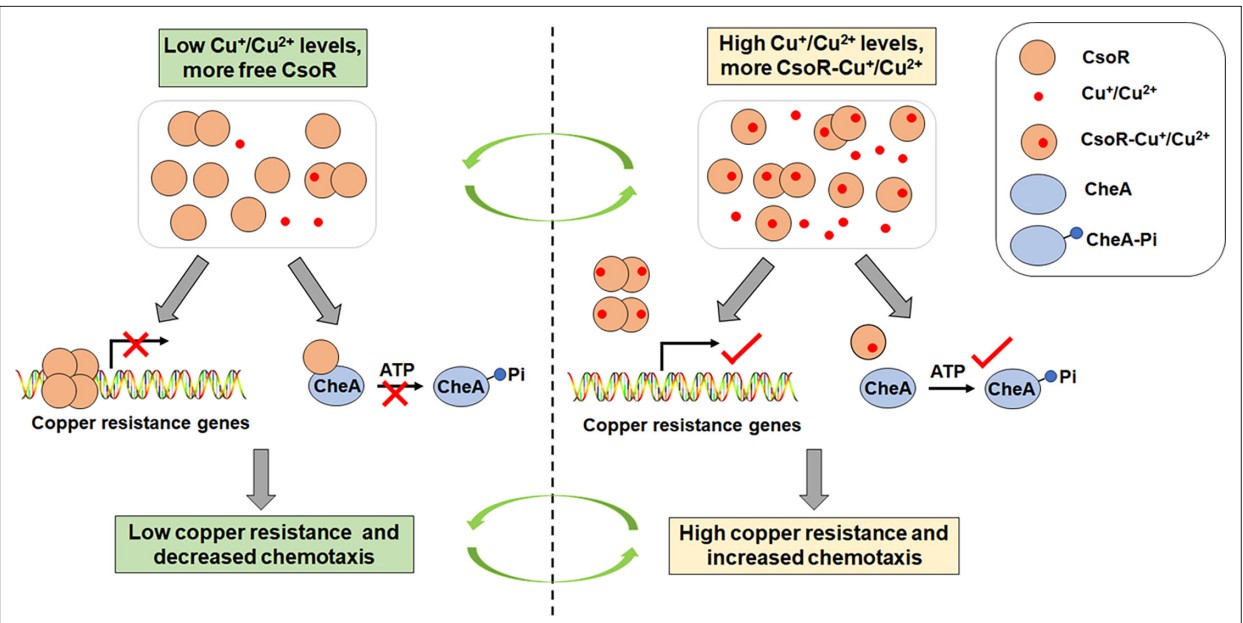

**Figure 8.** A proposed model for describing how CsoR coordinates chemotaxis and resistance to copper in *P.putida*. Under low Cu$^+$/Cu$^{2+}$ levels, more none Cu$^+$/Cu$^{2+}$-binding CsoR molecules (free CsoR) exist in the cell, and the free CsoR forms tetramer and binds to promoters of copper-resistance genes (such as *copA-I* and *copA-II*), leading to repressed gene transcription and low copper resistance. Meanwhile, free CsoR interacts with CheA and inhibits its autophosphorylation activity, decreasing chemotaxis ability. In contrast, more Cu$^+$/Cu$^{2+}$-binding CsoR molecules exist under high Cu$^+$/Cu$^{2+}$ levels, and the Cu$^+$/Cu$^{2+}$-binding changes the conformation of the CsoR tetramer and releases CsoR from target promoters, leading to increased gene transcription and copper resistance. Besides, Cu$^+$/Cu$^{2+}$-binding of CsoR decreases the interaction between CsoR and CheA, which relieves the inhibition of CsoR on CheA autophosphorylation, resulting in increased chemotaxis ability.

revealed that CsoR inhibited bacterial chemotaxis via interacting with CheA and hindering its auto-phosphorylation. Meanwhile, CsoR regulated copper resistance by modulating the expression of copper-resistance genes. Together with previous reports (*Chang et al., 2014*; *Jacobs et al., 2015*; *Tan et al., 2014*), we proposed a potential model to describe the function of CsoR in regulating copper resistance and bacterial chemotaxis. As shown in *Figure 8*, under low copper levels, CsoR molecules exist mainly in none copper-binding status (free CsoR) in the cell, and the free CsoR forms tetramer and binds to promoters of copper-resistance genes (such as *copA-I*), leading to repressed gene transcription and copper resistance ability. Meanwhile, the free CsoR interacts with CheA and inhibits its autophosphorylation, decreasing bacterial chemotaxis ability. Under high copper levels, more copper-binding CsoR molecules exist, and copper-binding changes the conformation of the CsoR tetramer and releases CsoR from promoters, leading to increased gene transcription and copper resistance. Besides, the copper-binding of CsoR decreases the interaction between CsoR and CheA, which relieves the inhibition of CsoR on CheA autophosphorylation, resulting in increased chemotaxis ability.

In classical chemotaxis signaling, CheA interacts with CheY, CheW, and CheB in the classical chemotaxis pathway. This study found 16 new CheA-interacting proteins using pull-down assay and subsequent analysis. Moreover, in another unpublished result, we found that CheA interacted with eight c-di-GMP-metabolizing proteins, and CheA transferred the phosphate group to one of them. Together, it seemed that CheA could interact with at least 27 proteins. With such a heterogeneous pool of CheA-complexes, performing a specific response seemed difficult. However, several previous studies have reported the example of one protein interacting with dozens of proteins. For example, the c-di-GMP effector LapD in *P. fluorescens* and *P. putida* can interact with a dozen different c-di-GMP-metabolizing proteins (*Giacalone et al., 2018*; *Nie et al., 2024*). In *E. coli*, a subset of DGCs and PDEs operated as central interaction hubs in a larger 'supermodule' by interacting with dozens of proteins (*Sarenko et al., 2017*). We infer that the expression of different CheA-interacting proteins might happen at different growth stages or under different conditions, and their interaction with

CheA under that stage/condition changed bacterial chemotaxis or the process in which the target protein was involved.

In classical chemotaxis signaling, MCP on the cell membrane senses external signaling molecules and regulates bacterial chemotaxis by mediating CheA autophosphorylation activity (*Porter et al., 2011*; *Ortega et al., 2017*). MCPs can directly bind diverse external signaling molecules, such as amino acids, dipeptides, sugars, tricarboxylic acid cycle intermediates, aromatic molecules, and inorganic phosphate (*Bi and Lai, 2015*). However, there are only a few reports on the relationship between MCP and metal ions (*Chandrashekhar et al., 2018*; *Li et al., 2022*; *Martín-Mora et al., 2016*), and no evidence supports that MCP senses metal ions by direct binding. It is possible that bacteria sense and trigger chemotaxis to metal ions differently from that they sense and trigger chemotaxis to external signaling molecules like amino acids. Our results provide a mechanism by which bacteria sense copper and regulate chemotaxis via the copper-responsive repressor CsoR. Through the interaction between CsoR and CheA, bacteria coordinately regulated chemotaxis and resistance to copper stress, which would favor the bacteria to better adapt to complex environments. Besides, the interaction between CsoR and CheA was not limited to the proteins from *P. putida*, and it was also found in proteins from several other bacterial species (*Figure 7*), implying that the regulation of chemotaxis and resistance to copper via the interaction between CsoR and CheA might be a widespread regulatory mechanism.

Although the *P. putida* CsoR functioned as a copper-responsive regulator to modulate the expression of copper-resistance genes, its effect on gene expression was much weaker than its homologous protein in other bacterial species. In *M. tuberculosis*, *B. subtilis*, *C. glutamicum*, *L. monocytogenes*, and *S. aureus*, deletion of *csoR* resulted in an about 10-fold increase in the expression of target genes in the absence of copper (*Marcus et al., 2016*; *Smaldone and Helmann, 2007*; *Teramoto et al., 2015*; *Corbett et al., 2011*; *Baker et al., 2011*). In contrast, deletion of *csoR* in *P. putida* led to a slight but reproducible increase (about 1.3-fold) in gene expression in the absence of copper (*Figure 4b*). This difference might be attributed to the existence of several key regulators that activated the expression of copper-resistance genes in response to copper in *P. putida*, such as CueR and CopR. CueR positively regulated the expression of *cueA*, encoding a copper-transporting P1-type ATPase that played a crucial role in copper resistance (*Adaikkalam and Swarup, 2002*). CopR was essential for expressing several genes implicated in cytoplasmic copper homeostasis, such as *copA-II*, *copB-II*, and *cusA* (*Quintana et al., 2017*; *Quaranta et al., 2009*). The existence of these positive regulators makes the function of CosR a secondary or even dispensable insurance in the expression of copper-resistance genes. Consistent with this, there is no CosR homolog in *P. aeruginosa*, and copper homeostasis in *P. aeruginosa* is mainly controlled by CueR and CopR (*Hofmann et al., 2021*; *Quintana et al., 2017*).

Through pull-down, BTH, and BiFC assays, we obtained 16 new CheA-interacting proteins involved in different physiological processes (*Supplementary file 1a*). Among the 16 proteins, 5 proteins (CsoR, IspG, NfuA, PhaD, and PP_1644) inhibited bacterial chemotaxis on semisolid plates (*Figure 2a*). Our study here focused on the physiological role of CsoR-CheA interaction. Still, the function of other interactions remained unclear. PhaD is a TetR family transcriptional regulator that behaves as a carbon source-dependent activator of the *pha* cluster related to polyhydroxyalkanoates (PHAs) biosynthesis (*de Eugenio et al., 2010*; *Tarazona et al., 2020*). Bacterial PHAs are isotactic polymers synthesized under unfavorable growth conditions in the presence of excess carbon sources. PHAs are critical in central metabolism, acting as dynamic carbon reservoirs and reducing equivalents (*Gregory et al., 2022*). The interaction between PhaD and CheA leads one to speculate that there might be some connection between PHA synthesis and bacterial chemotaxis. Another CheA-interacting protein, PP_1644, also attracts our interest. PP_1644 is annotated as a NAD(P)H dehydrogenase involved in cyclic electron transport and respiration processes. Exploring the physiological role of the interaction between CheA and these proteins in the future helps to reveal the association between the chemotaxis process and other physiological metabolisms.

## Materials and methods
### Bacterial strains and growth conditions
All strains and plasmids used in this study are listed in *Supplementary file 1b*. Unless specifically mentioned, *E. coli* strains were grown in lysogeny broth (LB) medium at 37 °C. *P. putida* KT2440

and its derivative strains were cultured at 28 °C in LB medium or chemically defined M9 minimal medium supplemented with 40 mM glucose as carbon source. Antibiotics were used, when required for plasmid maintenance or transformants screening, at the following concentrations: kanamycin (50 mg/L), carbenicillin (50 mg/L), chloramphenicol (25 mg/L), and gentamycin (20 mg/L for *E. coli* or 40 mg/L for *P. putida*).

## Plasmid and strain construction

The routine cloning of DNA fragments into plasmid was performed by following a T5 exonuclease-dependent method (*Xia et al., 2019*). Briefly, a 14 base pairs (bp) homologous end was added to the 5' of primer during synthesizing. Amplified DNA fragments and linearized plasmid containing the same homologous end were incubated in a reaction buffer containing T5 exonuclease (0.04 U) and left at 30°C for 40 min before transforming into *E. coli* competent cells. Primers used for plasmid construction are listed in *Supplementary file 1c*. All cloning steps involving PCR were verified by commercial sequencing (Tsingke, Wuhan, China).

Gene deletion mutant was constructed by homologous recombination using the suicide plasmid pBBR401. For example, to construct a markerless *P. putida csoR* deletion mutant, ~800 bp from the chromosomal regions flanking *csoR* (upstream region and downstream region) were PCR-amplified. The PCR products were cloned into pBBR401 to create pBBR401-*csoR*UP-DOWN. Then, the final plasmid was transferred to *P. putida* by electroporation. The integration strain was selected on plates containing gentamicin. After subculturing the integration strain in LB medium without antibiotics six times (12 hr each time), single colonies were obtained by plate streaking. Then, colonies losing gentamicin resistance were kept for further verification. The *csoR* delete mutant was confirmed by PCR and sequencing.

To generate a *csoR/cheA* overexpression plasmid, a DNA fragment containing the complete *csoR/cheA* was PCR amplified. The product was cloned into expressional vector pBBR403 to yield pBBR403-*csoR/cheA*. The expression of *csoR/cheA* on pBBR403-*csoR/cheA* was controlled by an inducible *tac* promoter. To construct a vector for target protein expression and purification, we amplified and cloned the target gene into pET-28a with 6×His tag or pHS-Strep with Strep II tag. Overlapping PCR was used to create point mutations of CsoR. To construct a C40A point mutation in CsoR, we amplified two fragments with two primer pairs (CsoR$_{C40A}$ s1/CsoR$_{C40A}$ a1 and CsoR$_{C40A}$ s2/CsoR$_{C40A}$ a2). The CsoR$_{C40A}$ a1 and the CsoR$_{C40A}$ s2 shared reverse complementary sequences containing the point mutation in which the original TGC codon of arginine was replaced by GCC of alanine. The two fragments were mixed in a 1:1 ratio to perform overlapping extension. The final PCR product was cloned into pET-28a and pBBR403. The mutation in *csoR* was confirmed by sequencing.

## Expression and purification of His/Strep II-tagged proteins

For the expression of His/Strep II-tagged protein, overnight culture of *E. coli* BL21 carrying the construct of target proteins was 1:100 diluted into LB medium and incubated for 4 hr at 37 °C. Then, 0.4 mM IPTG (isopropyl-D-thiogalactopyranoside) was added to induce protein expression. After 4 hr incubation at 16 °C, cells were harvested and resuspended in lysing buffer (10 mM Tris-Cl [pH 7.8], 300 mM KCl, and 10% (w/v) glycerol). The harvested cells were lysed using a pressure cell breaking apparatus, and cell debris was removed by centrifugation at 15,000 rpm for 20 min. The supernatants were then filtered through a 0.22-µm-pore-size filter and loaded onto a Ni-NTA Resin column (for His-tagged protein) or Strep-Tactin Resin column (for Strep II-tagged protein). Target proteins were eluted using an imidazole gradient (50/100/150/250 mM imidazole for His-tagged protein) or 5 mM biotin (for Strep II-tagged protein) and then dialyzed overnight against lysing buffer to remove imidazole. The concentrations of obtained proteins were determined using BCA assay.

## Protein-protein pull-down assay

Protein-protein pull-down assay was used to identify CheA-interacting protein and test the effect of copper on CheA-CsoR interaction. Briefly, 6×His/Strep II-tagged CheA was induced and loaded to a Ni-NTA/Strep-Tactin column as described above. Then, overnight cultured wild-type KT2440/BL21 strain expressing CsoR was harvested, lysed, and filtered before adding to the same column. For the pull-down assay to identify CheA-interacting protein, the same volume of wild-type extract was added to a blank Ni-NTA column as a negative control. For the pull-down assay to test the effect of copper

on CheA-CsoR interaction, various amounts of $CuCl_2$ were mixed with the cell extract containing CsoR before being added to the Strep-Tactin column. Then, the columns were sealed and incubated at 4 °C with 40 rpm shaking. After 2 hr incubation, the supernatant was removed, and the columns were washed with lysing buffer containing 20 mM imidazole (for Ni-NTA column) or lysing buffer (for Strep-Tactin column) to wash the unspecific binding protein away. Then, elution buffer containing 250 mM imidazole (for the Ni-NTA column) or 5 mM biotin (for the Strep-Tactin column) was added to wash down all proteins on the column. The eluted proteins were collected and resolved by 12.5% SDS–PAGE followed by Coomassie blue staining and mass spectrometry analyses.

## Mass spectrometry (MS)-based protein sequencing

After Coomassie blue staining of eluted proteins obtained from pull-down assay, the whole lane of the experimental or control sample was excised from gels and prepared for MS analysis. Protein from the excised gels was extracted with a Micro Protein PAGE Recovery Kit (Sangon Biotech, China) following the operating instructions. Then, trypsin digestion of extracted protein was performed with 1 g trypsin (Promega, USA) and incubated at 37 °C overnight. The digestion was terminated by adding trifluoroacetic acid (TFA). Then, desalting was subsequently performed using Zip-tip (Merck Millipore, Ireland). Peptides were eluted from the Zip-tip with 50 µL of matrix solution (5 mg/mL α-cyano-4-hydroxycinnamic acid, 50% acetonitrile, 0.1% TFA). The supernatant was collected and concentrated to a final volume of 10 µL in a centrifugal concentrator.

The samples were analyzed using the MALDI-TOF/TOF mass spectrometer (Applied Biosystems, USA). Mass spectra were recorded in the positive-ion mode, averaging 2500 laser shots per spectrum. Mass spectra (excluding trypsin autolytic peptides and other known background ions) were searched against the *P. putida* proteome from the UniProt database to identify the proteins. The search was performed using trypsin digestion, allowing two missed cleavages, specifying carbamidomethyl-Cys as a fixed modification, and setting a peptide mass tolerance of ±1.6 Da. The global false discovery rate (FDR) cutoff for peptide and protein identification was set to 0.01. An intensity-based absolute quantification (iBAQ) algorithm was used to rank the relative abundance of different proteins as previously described (*Schwanhäusser et al., 2011*). iBAQ percentage of specific proteins in the experimental sample (iBAQ_T (%)) and control sample (iBAQ_CK (%)) were used to represent relative protein concertation. $Log_2$ (iBAQ_T/iBAQ_CK) fold change of ≥2 or ≤-2 and a p value of≤0.05 was considered significantly different.

## Bacterial two-hybrid (BTH) assay

For bacterial two-hybrid analysis of protein-protein interactions of *P. putida* proteins expressed in *E. coli*, each ORF was cloned in-frame with the T18 and T25 fragments of adenylate cyclase ORF in vectors pUT18C and pKT25. Primers used to amplify each ORF are listed in *Supplementary file 1c*. The resulting vectors were co-transformed into *E. coli* BTH101 and plated onto LB agar plate supplemented with 50 mg/L carbenicillin, 50 mg/L kanamycin, 40 mg/L 5-bromo-4-chloro-3-indolyl-b-D-galactopyranoside (X-gal), and 0.5 mM IPTG, and incubated at 28 °C for 48 hr. Three co-transformants for each assay were cultured to stationary phase in LB broth at 28 °C, then spotted onto an LB agar plate supplemented as above, and incubated for 60 hr at 28 °C. Plates were then photoed on a Tanon 2500 scanner. After the photographs were taken, the colonies on the plates were scraped off, and the LacZ activities of obtained cells were measured using o-nitrophenyl-β-galactopyranoside (ONPG) as substrate, as described before (*Schaefer et al., 2016*). The experiments were repeated three times with three technical repeats per sample, and the data are presented as Miller units.

## Bimolecular fluorescence complementation (BiFC) assay

BiFC was used to analyze protein-protein interactions as previously described (*Chu et al., 2009*). Briefly, to determine the interaction between two interested proteins (such as CheA and CsoR), CheA was fused to KN151 (the N-terminal of mLumin), and CsoR was fused to LC151 (the C-terminal of mLumin), yielding a recombinant plasmid pBBR403-CheA-KN151-CsoR-LC151. Then, the recombinant plasmid was transformed into the wild-type *P. putida* strain. Transformants were picked and cultured in LB medium containing 40 mg/L gentamycin and 0.5 mM IPTG for 24 h at 28°C. Then, images of the dark-filed and bright field of the transformant cells were obtained using FV1000 CLSM (Olympus, Japan) equipped with a 100×/1.4 oil immersion objective lens. Besides, the transformant

cells were washed twice with 0.9% NaCl, and then fluorescence intensities and $OD_{600}$ were measured using a Spark microplate reader (Tecan, Switzerland). Fluorescence intensity and $OD_{600}$ were detected using the black and transparent microplate, respectively. The excitation and the emission wavelength to detect fluorescence were 587 nm and 620 nm, respectively, and the experiment was repeated twice with triplicates.

## Microscale thermophoresis (MST) assay

MST was performed to analyze the interaction between two proteins or proteins and metal ions. Briefly, to test the interaction between CsoR and copper. A green fluorescent protein (GFP) encoding gene was fused to the end of *csoR* in pET-28a-*csoR* to achieve fusion expression. The fusion protein CsoR-GFP was induced and purified as described above in protein induction and purification. The obtained CsoR-GFP was dialyzed with MST buffer (50 mM Tris-HCl [pH 7.8], 150 mM NaCl, 0.05% Tween 20). The MST assay was performed on a Monolith Instrument NT.115 device using standard treated capillaries (NanoTemper Technologies, Germany). The concentration of CsoR-GFP was constant at 250 nM, and the $CuCl_2$ concentration was varied from 0.031 to 1000 µM with a twofold gradient. The experiment was recorded using the Nano-BLUE fluorescent detector. Measurements were performed in MST buffer. The MO. Affinity Analysis software (version 2.3) was used to calculate the dissociation constant ($K_d$) from triplicate reads of measurements.

## In vitro phosphorylation assays

For the autophosphorylation reaction, purified CheA (3 µM) was incubated in phosphorylation buffer containing 50 mM Tris–HCl [pH 7.5], 15 mM $MgCl_2$, and 50 mM NaCl. The reaction was initiated by adding 0.03 µCi of [$^{32}$P]ATP[γP] (PerkinElmer, USA) to the mixture. SDS-PAGE loading buffer containing SDS and EDTA was added to the mixture to terminate the reaction at the indicated time. To test the effect of target proteins on CheA autophosphorylation, target protein (10 µM) was mixed with CheA (3 µM) for 10 min before adding [$^{32}$P]ATP[γP]. To test the transphosphorylation reaction, CheA was autophosphorylated before CheY/target protein (10 µM) was added to the mixture, and the reaction mixture was incubated at 30 °C for different time intervals before being terminated with SDS-PAGE loading buffer. Samples were heated at 95 °C for 5 min and then resolved by 12.5% SDS–PAGE. After drying of the gels, products were visualized by autoradiography.

## RNA extraction and real-time RT-PCR (qRT-PCR) assay

*P. putida* cells were cultured in M9 minimal medium supplemented with 40 mM glucose as carbon source for 24 hr. Then, cells were harvested and washed thrice with sterilized phosphate buffer saline (PBS) before being divided into two equal parts. One part was resolved with fresh M9 medium and another with M9 medium containing 10 µM $CuCl_2$. After 30 min incubation, cells were harvested for RNA extraction using a total RNA extraction reagent (Vazyme R401-01, China) as recommended by the manufacturer. 1 µg extracted RNA was digested with DNase I and reverse transcribed to cDNA using a reverse transcription kit (Takara RR047A, Japan), and cDNA was used as the template for qRT-PCR analysis. The qRT-PCR assay was performed using Power SYBRTM Green PCR mix (Applied Biosystems 4367659, USA) and analyzed using a QuantStudio 3 Real-Time PCR System (Applied Biosystems, USA). The *rpoD* gene was selected as an internal control. The primers used in qRT-PCR analysis are listed in *Supplementary file 1c*. All experiments were performed thrice with three technical repeats per sample.

## Electrophoretic mobility shift assay (EMSA)

EMSA was used to test the interaction between CsoR and *copA-I* promoter DNA. Equal amounts of DNA (60 ng) were added to binding reactions with various quantities of CsoR in binding buffer (10 mM Tris-Cl [pH 7.8], 50 mM KCl, 20 mM $MgCl_2$, 5% glycerol, 20 µL total reaction volume). CsoR was incubated with promoter DNA for 20 min at room temperature. Reaction mixtures containing $Cu^{2+}$/DTT/$Cu^{2+}$+DTT were performed as described above, except $Cu^{2+}$/DTT/$Cu^{2+}$+DTT was incubated with CsoR for 10 min before adding DNA. All reaction solutions were loaded onto 5% acrylamide gel and electrophoresed at 150 V for 40 min in 0.5×TBE buffer (45 mM Tris-Cl [pH 7.8], 45 mM borate, 1 mM EDTA). Gels were stained with ethidium bromide before being digitized using a scanner (Tanon 2500, China).

## Bacterial chemotaxis assay

The chemotaxis ability of *P. putida* strains was assessed by using semisolid plate and μ-slide Chemotaxis plate (Ibidi 80326, Germany). For the method with semisolid plates, an agar plug (1% agar) containing 200 mM $CuCl_2$ was placed in the center of a LB semisolid plate (0.25% agar) and left at room temperature for 12 hr to achieve a $CuCl_2$ gradient on the semisolid plate. An agar plug without $CuCl_2$ was also placed in the center of a semisolid plate and used to test bacterial chemotaxis without copper. For the assay to investigate the effect of copper on bacterial chemotaxis, 200 μM $CuCl_2$ (final concentration) was mixed with semisolid LB before making a semisolid plate. Overnight growth *P. putida* cells were washed and resuspended with fresh M9 medium and adjusted to the same optical density ($OD_{600}$=0.5). Then, 2 μL of resuspended cultures was spotted 2 cm away from the agar plug/plate center, and the plate was incubated for 16 hr (for the control plate) or 18 hr (for the copper-containing plate) at 28 °C before digital photographs were taken. The distance from the point of inoculation to the edge of the colony growth closest to the agar plug (D1) and the distance from the point of inoculation to the edge of the colony growth farthest from the agar plug (D2) were measured. The response index (RI) value was calculated to characterize the bacterial response to $CuCl_2$ as previously described (*Pham and Parkinson, 2011*). The RI was calculated using the following equation: RI = D1/(D1 + D2). RI values greater than 0.52 and less than 0.48 correspond to attractant and repellent responses, and intermediate values represent nonresponses.

For the method with μ-slide Chemotaxis plate (a commercially available microfluidic device with a channel connecting two reservoirs), the reservoirs were filled with the two bacterial solutions (with or without $CuCl_2$) following a modified version of the manufacturer's "Fast Method" protocol. Briefly, overnight cultures were inoculated with M9 medium and grown at 28 °C, 180 rpm, until they reached an optical density ($OD_{600}$) between 0.3 and 0.35. 1 ml of bacterial culture was washed twice with fresh M9 medium by centrifugation (3 min at 3000 rpm), and finally diluted to a target $OD_{600}$ of 0.015 with fresh M9 supplemented with 0.01% Tween 20, with or without 2 mM $CuCl_2$ for injection into the chemotaxis device. First, the entire device was overfilled with buffer free of $CuCl_2$ or bacteria through the filling ports, and then the central channel's ports were closed with plugs. About 65 μl was removed from one reservoir, replaced by 65 μl of $CuCl_2$-free bacterial solution, and then this reservoir's ports were closed. Finally, all liquid was removed from the other reservoir and replaced with a bacterial solution containing $CuCl_2$. For control measurements, neither bacterial solution contained $CuCl_2$. Phase contrast microscopy recordings were obtained at room temperature on a Nikon Ti-E inverted microscope using an sCMOS camera (PCO Edge 4.2) and a ×40 objective lens. Recordings were obtained starting from 20 min after filling the device. Three 15 s-long recordings of cells in the observation area were obtained at 15 fps. Then, single-cell tracking analysis was performed using ImageJ's Manual Tracking and Chemotaxis Tool plugins.

## Statistical analysis

Statistical analyses were performed using Graph-Pad Prism (version 9.0.0). Student's t-test or an ANOVA was used to analyze the significance of differences in LacZ activity, fluorescence intensity, swimming zone diameter, gene expression, response index, center of mass, and velocity. A p value less than 0.05 was considered statistically significant.

## Acknowledgements

This work was supported by the National Key Research and Development Program of China [2020YFC1806803], the Hubei Provincial Natural Science Foundation of China [2024AFB691], the National Natural Science Foundation of China [31900054], and the Fundamental Research Funds for the Central Universities [2662022SKQD002].

## Additional information

### Funding

| Funder | Grant reference number | Author |
|---|---|---|
| National Key Research and Development Program of China | 2020YFC1806803 | Qiaoyun Huang |
| Hubei Provincial Natural Science Foundation of China | 2024AFB691 | Yujie Xiao |
| National Natural Science Foundation of China | 31900054 | Yujie Xiao |
| Fundamental Research Funds for the Central Universities | 2662022SKQD002 | Yujie Xiao |

The funders had no role in study design, data collection and interpretation, or the decision to submit the work for publication.

### Author contributions

Meina He, Data curation, Software, Formal analysis, Investigation, Writing - original draft; Yongxin Tao, Haoqi Feng, Ying Fan, Tong Liu, Investigation, Methodology; Kexin Mu, Validation, Investigation; Qiaoyun Huang, Conceptualization, Supervision, Funding acquisition; Yujie Xiao, Conceptualization, Resources, Formal analysis, Supervision, Funding acquisition, Writing – review and editing; Wenli Chen, Conceptualization, Supervision, Funding acquisition, Writing – review and editing

### Author ORCIDs

Wenli Chen (ID) https://orcid.org/0000-0003-1717-1263

Reviewer #2 (Public review): https://doi.org/10.7554/eLife.100914.3.sa1
Author response https://doi.org/10.7554/eLife.100914.3.sa2

## Additional files

### Supplementary files

Supplementary file 1. Tables containing information for target proteins, strains, plasmids, and primers in this work. (a) Target proteins identified in the pull-down assay. (b) Strains and plasmids used in this work. (c) Primers used in this work.

MDAR checklist

### Data availability

All data generated or analysed during this study are included in the manuscript and supporting files. The original gels/blots generated in this study have been provided in the source data files. Relevant data supporting the critical findings of this study are available within the article and the Supplementary file. The materials used in this study are available upon reasonable request.

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
