## [Editor Report · eLife Assessment]

Data presented in this **useful** report suggest a potentially new model for chemotaxis regulation in the gram-negative bacterium P. putida. Data supporting interactions between CheA and the copper-binding protein CsoR, reveal potential mechanisms for coordinating chemotaxis and copper resistance. There was, however, concern about the large number of CheA interactors identified in the initial screen and it was felt that the study was **incomplete** without a substantial number of additional experiments to test the model and bolster the authors' conclusions.

---

## [Referee Report · Reviewer #2 (Public review)]

Summary:

This manuscript focuses on the apparent involvement of a proposed copper-responsive regulator in the chemotactic response of Pseudomonas putida to Cu(II), a chemorepellent. Broadly, this area is of interest because it could provide insight into how soil microbes mitigate metal stress. Additionally, copper has some historical agricultural use as an antimicrobial, thus can accumulate in soil. The manuscript bases its conclusions on an *in vitro* screen to identify interacting partners of CheA, an essential kinase in the P. putida chemotaxis-signaling pathway. Much of the subsequent analysis focuses on a regulator of the CsoR/RcnR family (PP_2969).

Weaknesses:

The data presented in this work does not support the model (Figure 8). In particular, PP_2969 is linked to Ni/Co resistance not Cu resistance. Further, it is not clear how the putative new interactions with CheA would be integrated into diverse responses to various chemoattract/repellents. These two comments are justified below.

PP_2969

• The authors present a sequence alignment (Figure S5) that is the sole based for their initial assignment of this ORF as a CsoR protein. There is conservation of the primary coordinating ligands (highlighted with asterisks) known to be involved in Cu(I) binding to CsoR (ref 31). There are some key differences, though, in residues immediately adjacent to the conserved Cys (the preceding Ala, which is Tyr in the other sequences). The effect of these change may be significant in a physiological context.

• The gene immediately downstream of PP_2969 is homologous to *E. coli* RcnA, a demonstrated Ni/Co efflux protein, suggesting that P2969 may be Ni or Co responsive. Indeed PP_2970 has previously been reported as Ni/Co responsive (J. Bact 2009 doi:10.1128/JB.00465-09). The host cytosol plays a critical role in determining metal-response, in addition to the protein, which can explain the divergence from the metal response expected from the alignment.

• The previous JBact study also explains the lack of an effect (Figure 5b) of deleting PP_2969 on copper-efflux gene expression (copA-I, copA-II, and copB-II) as these are regulated by CueR not PP_2969 consistent with the previous report. Deletion of CsoR/RcnR family regulator will result in constitutive expression of the relevant efflux/detoxification gene, at a level generally equivalent to the de-repression observed in the presence of the signal.

• Further, CsoR proteins are Cu(I) responsive so measuring Cu(II) binding affinity is not physiologically relevant (Figures 5a and S5b). The affinities of demonstrated CsoR proteins are 10-18 M and these values are determined by competition assay. The MTS assay and resulting affinities are not physiologically relevant.

• The DNA-binding assays are carried out at protein concentrations well above physiological ranges (Figs 5c and d, and S5c, d). The weak binding will in part result from using DNA-sequences upstream of the copA genes and not from from PP_2970.

CheA interactions

There is no consideration given to the likely physiological relevance of the new interacting partners for CheA.

• How much CheA is present in the cell (copies) and how many copies of other proteins are present? How would specific responses involving individual interacting partners be possible with such a heterogenous pool of putative CheA-complexes in a cell. For PP_2969, the affinity reported (Figure 5A) may lay at the upper end of the CsoR concentration range (for example, CueR in *Salmonella* is present at ~40 nM).

• The two-hybrid system experiment uses a long growth time (60 h) before analysis. Even low LacZ activity levels will generate a blue colour, depending upon growth medium (see doi: 10.1016/0076-6879(91)04011-c). It is also not clear how Miller units can be accurately or precisely determined from a solid plate assay (the reference cited describes a protocol for liquid culture).

Comments on revised version:

The authors have replied in detail to the various comments about the original manuscripts. However, the responses are generally lengthy rationalisations of the original interpretation of the data and do not fundamentally address critical concerns raised about the physiological relevance of the results. The response appears to rest on the assumption that the numerous interacting partners obtained from the initial screen are all true positives and that all subsequent experimental results are interpreted to justify that assumption. In the case of CsoR, the experimental results and interpretation are inconsistent with previously published studies of the metal and DNA-binding properties of CsoR proteins. The following points reiterate comments from the previous review, in the hopes that the authors will, at the very least, consider the likelihood that the "CsoR" protein they have identified is in fact responsive to a different metal. Further, that the authors consider multiple possible interpretations of the data, particularly those that are inconsistent with the model/hypothesis and take this into account in their experimental design.

• (Figure 4) Almost all purified proteins will bind Cu(II) most tightly in vitro, followed by Zn(II) and Ni(II). This behaviour is a consequence of the Irving-Williams affinity series (doi.org/10.1038/162746a0 and doi.org/10.1039/JR9530003192, especially Figure 4) and is not considered an indicator of physiological metal preference. Biomolecules will exhibit the same behaviour as small organic ligands towards first row transition ions because of the flexibility of their structures. Thus, the results obtained are unsurprising and, because of the method used, have no physiological relevance.

• The authors cite other in vivo work as evidence for varied metal-response by regulator proteins. However, experiments in these citations are of limited relevance because some focus on other structural classes of metalloregulator proteins (so not relevant here) while others focus on changes in metal accumulation by overexpression of the regulator protein, with no examination of the metal-specificity of the efflux protein the key determinant of the physiological response of the regulator protein - why turn on expression of an efflux protein that can't pump out a particular metal? Finally, adding equivalent concentrations of metals to growing cells is not a good comparison as metals are toxic at different concentrations. The regulators will only have evolved to be just good enough, not perfect, with respect to selectivity. Laboratory experimental conditions often explore non-physiological conditions.

• It is also important to re-emphasise the authors' own statements on lines 90-93 that P. putida has a CueR protein. This is consistent with the phylogenetic distribution of CueR proteins in gram-negative bacteria. The CsoR proteins, in contrast, are found only in gram-positive bacteria. This inconsistency is ignored by the authors.

• The implications of the Irving-Williams series on metal-specific responses of bacterial metalloregulator proteins are described in the following references: 10.1016/j.cbpa.2021.102095, 10.1074/jbc.R114.588145, and 10.1038/s41589-018-0211-4. The last reference of this set provides an experimental basis for why metalloregulator affinities for Cu (and Zn and Ni) are so tight (and why the values obtained in Figure 4 in this manuscript are not relevant).

• Similarly, the previous experimental studies of CsoR proteins not cited by the authors (10.1021/ja908372b 10.1021/bi900115w) provide rigourous experimental approaches for measuring metal and DNA-binding affinities and further highlight the weakness of the experimental design in this manuscript.

• The DNA-binding assays are not physiologically relevant because they do not use DNA from the operator regulated by the candidate protein (why this was not explored in the revision is difficult to understand). The mobility shift observed at these high protein concentrations will result from non-specific binding. It is unsurprising that Cu(II) has an effect on DNA binding as it is added at such high concentrations relative to both protein and DNA so as to compete for DNA-binding with the protein (which binds weakly because there is no specific recognition site). The 10:1 ratio of Cu:CsoR is 10-times higher than needed as this class of proteins will show decreases in DNA-affinity in the presence of the correct metal at 1:1 stoichiometry. As indicated above, the authors need to consider alternative interpretations for their results rather than try to rationalise the results to fit the model.

The points raised above readily address the authors' own comments in the response as to their surprise at some of the results and their inconsistency with the model.

Even if the authors were to identify the correct metal to which the protein responds, there are still fundamental issues with experimental design and interpretation that would need to be addressed to indicate any link between the protein and chemotaxis.

---

## [Author Response]

The following is the authors’ response to the original reviews.

**Reviewer #1 (Public Review):**
This report contains two parts. In the first part, several experiments were carried out to show that CsoR binds to CheA, inhibits CheA phosphorylation, and impairs P. putida chemotaxis. The second part provides some evidence that CsoR is a copper-binding protein, binds to CheA in a copper-dependent manner, and regulates P. putida response to copper, a chemorepellent. Based on these results, a working model is proposed to describe how CsoR coordinates chemotaxis and resistance to copper in P. putida. While the second part of the study is relatively solid, there are some major concerns about the first part.Critiques:(1) The rigor from prior research is not clear. In addition to talking about other bacterial chemotaxis, the Introduction should briefly summarize previous work on P. putida chemotaxis and copper resistance.

We summarized previous results on *P. putida* copper resistance and added those results to the introduction section of the revised manuscript. As for chemotaxis, most studies in *P. putida* focused on the sensing/responding of the bacteria to different chemical compounds and the methyl-accepting chemotaxis proteins (MCPs) involved in the sensing, which is not relevant to the main content of this study. The component of the chemotaxis system in *P. putida* is similar to that in *E. coli*, and the signaling mechanism is presumably similar.

(2) The rationale for identifying those CheA-binding proteins is vague. CheA has been extensively studied and its functional domains (P1 to P5) have been well characterized. Compared to its counterparts from other bacteria, does P. putida CheA contain a unique motif or domain? Does CsoR bind to other bacterial CheAs or only to P. putida CheA?

The original purpose of the pull-down assay was to detect the interaction between CheA and c-di-GMP metabolizing enzymes, which was another project. However, we ignored that most c-di-GMP metabolizing enzymes were membrane proteins, and we made a mistake by using whole-cell lysate in the pull-down experiment. Thus, we failed to identify c-di-GMP metabolizing enzymes in “target” proteins of the pull-down assay. However, we found several novel “target” proteins in the pull-down assay. We wondered about the function of these proteins and the physiological roles of the interaction between CheA and these proteins, which was the primary purpose of this study. Although the function of CheA has been well characterized, most previous results focused on the role of CheA in chemotaxis, and its role in other bacterial processes was poorly studied. To extend our knowledge about CheA, we analyzed the results of the pull-down assay and decided to test the interaction between CheA and identified proteins, as well as the physiological roles of the interaction.

BLAST results showed that the CheA of *P. putida* shared 41.12% sequence similarity with the CheA of *E. coli*, and the CheA of *P. putida* had a similar domain pattern to those CheAs from other bacteria. To test whether CsoR_P. putida_ interacted with CheA from other bacteria, we performed a BTH assay to investigate the interaction between CsoR_P. putida_ and eight CheAs, including CheA from *E. coli*, CheA from *A. caldus*, CheA from *B. diazoefficiens*, CheA from *B. subtilis*, CheA from *L. monocytogenes*, CheA from *P. fluorescens*, CheA from *P. syringae*, and CheA from *P. stutzeri*. As shown in the following Fig. 1, CsoR_P. putida_ could interact with CheA from *A. caldus*, *B. subtilis*, *L. monocytogenes*, *P. fluorescens*, *P. syringae*, and *P. stutzeri*. Besides, among these strains, *cheA* and *csoR* coexist in *A. caldus*, *B. diazoefficiens*, *B. subtilis*, *L. monocytogenes*, *P. fluorescens*, *P. syringae*, and *P. stutzeri*. We previously tested the interaction of the two proteins from these bacterial species. The results showed that the CheA-CsoR interaction existed between proteins from *A. caldus*, *B. subtilis*, *P. syringae*, and *P. stutzeri* (Fig. 7 in the manuscript). However, CheA and CsoR from *B. diazoefficiens*, *L. monocytogenes*, and *P. fluorescens* showed no apparent interaction (Fig. 7 in the manuscript). These results suggested that unique amino acid sequences in the two proteins might be required to achieve interaction.

(3) Line 133-136, "Collectively, using pull-down, BTH, and BiFC assays, we identified 16 new CheA-interacting proteins in P. putida." It is surprising that so many proteins were identified but none of them were chemotaxis proteins, in particular those known to interact with CheA, such as CheW, CheY and CheZ, which raises a concern about the specificity of these methods. BTH and BiFC often give false-positive results and thus should be substantiated by other approaches such as co-IP, surface plasmon resonance (SPR), or isothermal titration calorimetry (ITC) along with mutagenesis studies.

The response regulator CheY and the phosphatase CheZ (two proteins known to be associated with CheA) were identified in the pull-down assay (Table S1), and the two proteins showed high Log_2_(fold change) values, indicating that they were obtained in the pull-down assay with high amount in the experimental group and low amount in the control group. Our study aimed to identify new CheA-interacting proteins; thus, the two proteins (CheY and CheZ) were not included in subsequent investigations. The CheA-interacting proteins were initially obtained through an *in vitro* assay (pull-down), followed by an *in vivo* assay (BTH and BiFC) to test the interaction further. Only proteins that showed positive results in all three assays were considered trustworthy CheA-interacting proteins and kept for further study.

(4) Line 147-149, "Fig. 2a, five strains (WT+pcsoR, WT+pispG, WT+pnfuA, WT+pphaD, and WT+pPP_1644) displayed smaller colony than the control strain (WT+pVec), indicating a weaker chemotaxis ability in these five strains." If copper is a chemorepellent, these strains should swim away from high concentrations of copper; thus, the sizes of colonies couldn't be used to measure this response. In the cited reference (reference 29), bacterial response to phenol was measured using a response index (RI).

Except for CsoR, the rest of the CheA-interacting proteins had no direct connection with copper and were involved in different processes (Table S1). A reasonable speculation is that these proteins involved in different processes can integrate signals from specific processes into chemotaxis by regulating CheA autophosphorylation, leading to better regulation of chemotaxis according to intracellular physiological state. We used semisolid nutrient agar plates to test and compare bacterial chemotaxis ability. In a fixed attractant/repellent gradient, chemokine, such as copper, can lead to two subpopulations traveling at different speeds, with the slower one being held back by the chemokinetic drift. In the case of semisolid plate migration, bacteria with chemotaxis ability formed large colonies by generating their gradient by consuming nutrients/producing toxic metabolic waste and following attractant/repellent gradients leading outward from the colony origin (Cremer et al., 2019. Nature 575:658–663). The observation of successive sharp circular bands (rings) progressing outward from the inoculation point was taken to confirm the chemotaxis genotype, and mutants without chemotaxis spread out uniformly and formed a small colony (Wolfe and Berg, PNAS. 1989, 86:6973-6977). In our experiment, we were unsure about the signals/chemokines of each target protein, so we could not design a fixed attractant/repellent gradient. Besides, all target proteins interacted with CheA, which is a crucial factor in chemotaxis, and we assume that these proteins would affect chemotaxis under overexpression conditions. Thus, we used semisolid nutrient plates to test and compare bacterial chemotaxis ability.

(5) Figures 2 and 3 show both CsoR and PhaD bind to CheA and inhibit CheA autophosphorylation. Do these two proteins share any sequence or structural similarity? Does PhaD also bind to copper? Otherwise, it is difficult to understand these results.

Thanks a lot. This is an enlightening comment. CsoR is a protein with a size of 10.8 kDa, and PhaD is 23.1 kDa. Because of the difference in size, we took it for granted that the two proteins were not similar. We recently compared their sequence on NCBI BLAST. Although both CsoR and PhaD are transcriptional regulators and interact with CheA, they have no significant sequence similarity. In terms of protein structure, we predicted their structures using AlphaFold. The results showed that CsoR consisted of three α-helixes and PhaD consisted of nine α-helixes (new Fig. S5a and S5b in the manuscript). We further compared their structure using Pymol but found no significant similarity between the two proteins (new Fig. S5c in the manuscript).

PhaD is a TetR family transcriptional regulator located adjacent to the genes involved in PHA biosynthesis, and it behaves as a carbon source-dependent activator of the *pha* cluster related to polyhydroxyalkanoates (PHAs) biosynthesis (de Eugenio et al., Environ Microbiol. 2010, 12:1591-1603; Tarazona et al., Environ Microbiol. 2020, 22:3922-3936). Bacterial PHAs are isotactic polymers synthesized under unfavorable growth conditions in the presence of excess carbon sources. PHAs are critical in central metabolism, acting as dynamic carbon reservoirs and reducing equivalents (Gregory et al., Trends Mol Med. 2022, 28:331-342). The interaction between PhaD and CheA leads us to speculate that there might be some connection between PHA synthesis and bacterial chemotaxis. For example, chemotaxis helps bacteria move towards specific carbon sources that favor PHA synthesis, and the interaction between PhaD and CheA weakens chemotaxis, causing bacteria to linger in areas rich in these carbon sources. This is an interesting hypothesis worth testing in the future.

(6) Line 195-196, "CsoR/PhaD had no apparent influence on the phosphate transfer between CheA and CheY". CheA controls bacterial chemotaxis through CheY phosphorylation. If this is true, how do CsoR and PhaD affect chemotaxis?

During the autophosphorylation assay, CheA was mixed with CsoR/PhaD and incubated for about 10 min before adding [^32^P]ATP[γP]. Thus, the effect of CsoR/PhaD on CheA autophosphorylation happened through the assay, and a significant inhibition effect was observed in the final result. Regarding transphosphorylation, CheA was mixed with ATP and incubated for about 30 min, at which time the autophosphorylation of CheA happened. Then, CsoR/PhaD and CheY were added to the phosphorylated CheA to investigate transphosphorylation. CsoR and PhaD affected chemotaxis via inhibiting CheA autophosphorylation, which was a crucial step in chemotaxis signaling, and the decrease in CheA autophosphorylation caused decreased chemotaxis.

(7) Figure 3 shows that CsoR/PhaD bind to CheA through P1, P3, and P4. This result is intriguing. All CheA proteins contain these three domains. If this is true, CsoR/PhaD should bind to other bacterial CheAs too. That said, this experiment is premature and needs to be confirmed by other approaches.

As replied to comment (2) above, we performed a BTH assay to investigate whether CsoR_P. putida_ interacts with CheA from other bacterial species. The results revealed that CsoR_P. putida_ interacted with CheA from *A. caldus*, *B. subtilis*, *L. monocytogenes*, *P. fluorescens*, *P. syringae*, and *P. stutzeri*, but not with CheA from *E. coli* and *B. diazoefficiens*. This result suggested that CheA-CsoR interaction required specific/unique amino acid sequence patterns in the two proteins, and similar domain composition alone was insufficient.

(8) Figure 5, does PhaD contain these three residues (C40, H65, and C69)? If not, how does PhaD inhibit CheA autophosphorylation and chemotactic response to copper?

No, there is no significant sequence similarity between PhaD and CsoR, and PhaD contains none of the three residues of CsoR (C40, H65, and C69). The size of the two proteins is also quite different (CsoR 10.8 kDa, PhaD 23.1 kDa). The structure alignment also revealed no apparent similarity between the predicted structures of PhaD and CsoR (new Fig. S5c in the manuscript). Nevertheless, CsoR and PhaD interacted with CheA through its P1, P3, and P4 domains. It is interesting how the two proteins interacted with CheA, but we currently have no answer.

(9) Does deletion of cosR or cheA have any impact on P. putida resistance to high concentrations of copper?

No, deletion of *cosR*/*cheA* had no noticeable impact on *P. putida*'s resistance to high concentrations of copper. We performed a growth assay to test the effect of CsoR and CheA on copper resistance under both liquid and solid medium conditions. The copper concentration was set at 0, 200, 500, 1000 μM. With the increase of copper concentration, the growth of bacteria was gradually inhibited, but the growth trends of *csoR* mutant, *cheA* mutant, and complementary strains were similar to that of the wild-type strain (new Fig. S6b and S6c in the manuscript). We speculated that this might be attributed to CsoR being a repressor and inhibiting gene expression in the absence of copper. When copper existed, the inhibitory effect of CsoR was relieved, which is the same as that in the *csoR* mutant. Besides, although deletion of *cosR* led to a slight increase (about 1.3-fold) in the expression of copper resistance genes (Fig. 4b in the manuscript), its effect on gene expression was much weaker than its homologous protein in other bacterial species. In *M. tuberculosis*, *B. subtilis*, *C. glutamicum*, *L. monocytogenes*, and *S. aureus*, deletion of *csoR* resulted in an about 10-fold increase in the expression of target genes in the absence of copper. This difference might be attributed to several vital regulators that activated the expression of copper-resistance genes in response to copper in *P. putida*, such as CueR and CopR (Adaikkalam and Swarup, Microbiology. 2002, 148:2857-2867; Hofmann et al., Int J Mol Sci, 2021, 22:2050; Quintana et al., J Biol Chem, 2017, 292:15691-15704). CueR positively regulated the expression of *cueA*, encoding a copper-transporting P1-type ATPase that played a crucial role in copper resistance. CopR was essential for expressing several genes implicated in cytoplasmic copper homeostasis, such as *copA-II*, *copB-II*, and *cusA*. The existence of these positive regulators makes the function of CosR a secondary or even dispensable insurance in the expression of copper-resistance genes. Consistent with this, there is no CosR homolog in *P. aeruginosa*, and copper homeostasis is mainly controlled by CueR and CopR.

**Reviewer #2 (Public Review):**
This manuscript focuses on the apparent involvement of a proposed copper-responsive regulator in the chemotactic response of Pseudomonas putida to Cu(II), a chemorepellent. Broadly, this area is of interest because it could provide insight into how soil microbes mitigate metal stress. Additionally, copper has some historical agricultural use as an antimicrobial, thus can accumulate in soil. The manuscript bases its conclusions on an in vitro screen to identify interacting partners of CheA, an essential kinase in the P. putida chemotaxis-signaling pathway. Much of the subsequent analysis focuses on a regulator of the CsoR/RcnR family (PP_2969).Weaknesses:The data presented in this work does not support the model (Figure 8). In particular, PP_2969 is linked to Ni/Co resistance, not Cu resistance. Further, it is not clear how the putative new interactions with CheA would be integrated into diverse responses to various chemoattract/repellents. These two comments are justified below.

Thanks a lot for all these comments. Before designing experiments to explore the function of PP_2969, we found three clues: (i) its sequence showed 38% similarity to the copper-responsive regulator CsoR of *M. tuberculosis*, and the three conserved amino acids essential for copper-binding were conserved in PP_2969; (ii) it located next to a Ni^2+^/Co^2+^ transporter (PP_2968) on the genome; (iii) a previous report revealed that PP_2969 (also named MreA) expression increased during metal stress, and overexpression of PP_2969 in *P. putida* and *E. coli* led to metal accumulation (Zn, Cd, and Cr) (Lunavat et al., Curr Microbiol. 2022, 79:142). These clues indicate that the function of PP_2969 is related to metal-binding, but it remains to be explored which metal(s) PP_2969 binds to. Thus, we played MST assay to test the interaction between PP_2969 and metals, including copper (Cu^2+^), zinc (Zn^2+^), nickel (Ni^2+^), cobalt (Co^2+^), cadmium (Cd^2+^), and magnesium (Mg^2+^). The result showed that PP_2969 was bound to three metal ions (Cu^2+^, Zn^2+^, Ni^2+^), and the binding to Cu^2+^ was the strongest. Besides, the EMSA assay revealed that Cu^2+^/Ni^2+^/Zn^2+^ inhibited the interaction between PP_2969 and promoter DNA, and Cu^2+^ showed the most substantial inhibitory effect at the same concentration. These results suggested that PP_2969 was mainly bound to Cu^2+^, followed by Zn^2+^ and Ni^2+^. To further test whether PP_2969 functioned as a metal-responsive repressor and which metal resistance was related to its target gene, we constructed a *PP_2969* deletion mutant and complementary strain and performed a qPCR assay to compare the expression of metal resistance-related genes. 14 metal-resistant-related genes were chosen as targets. The results showed that *PP_2969* deletion led to a weak but significant increase (about 1.3-fold) in expression of 10 genes, including three copper-resistance genes (*copA-I*, *copA-II,* and *copB-II*), one nickel-resistance gene (*nikB*), two cadmium-resistance genes (*cadA-I* and *cadA-III*), one cobalt-resistance gene (*cbtA*), and three multiple metal-resistance genes (*czcC-I*, *czcB-II*, and *PP_0026*) (Fig. 4b, Fig. S5a in the manuscript). Meanwhile, complementation with a multicopy plasmid containing the *PP_2969* gene decreased the gene expression in Δ*PP_2969*. Although PP_2969 regulated the expression of multiple metal resistance genes, it showed the most robust binding to Cu^2+^. Thus, we considered its primary function as a Cu^2+^-responsive regulator.

As for the second comment, “How would the putative new interactions with CheA be integrated into diverse responses to various chemoattract/repellents?”, We have some speculations based on our results and previous reports. For example, PP_2969 interacted with CheA and decreased its autophosphorylation activity, and copper inhibited the interaction between CheA and PP_2969. In the absence of copper, PP_2969 binds to promoters to inhibit the expression of copper resistance genes, and it also binds to CheA to inhibit its autophosphorylation, resulting in lower chemotaxis. When the bacteria move to an area of high copper concentration, PP_2969 binds to copper and falls off the DNA promoter, leading to higher expression of copper resistance genes. Meanwhile, copper-binding of PP_2969 decreases its interaction with CheA, increasing CheA autophosphorylation promoting chemotaxis, and bacteria swim away from the high copper concentration. Another attractive target protein is PhaD, a TetR family transcriptional regulator located adjacent to the genes involved in PHA biosynthesis, and it behaves as a carbon source-dependent activator of the *pha* cluster related to polyhydroxyalkanoates (PHAs) biosynthesis (de Eugenio et al., Environ Microbiol. 2010, 12:1591-1603; Tarazona et al., Environ Microbiol. 2020, 22:3922-3936). Bacterial PHAs are isotactic polymers synthesized under unfavorable growth conditions in the presence of excess carbon sources. PHAs are critical in central metabolism, acting as dynamic carbon reservoirs and reducing equivalents (Gregory et al., Trends Mol Med. 2022, 28:331-342). The interaction between PhaD and CheA leads us to speculate that there might be some connection between PHA synthesis and bacterial chemotaxis. For example, chemotaxis helps bacteria move towards particular carbon sources that favor PHA synthesis; the regulator PhaD activates the genes related to PHA synthesis. Meanwhile, the interaction between PhaD and CheA weakens chemotaxis, causing bacteria to linger in areas rich in these carbon sources. Collectively, we speculate that by interacting with CheA and modulating its autophosphorylation, target proteins such as CsoR/PhaD integrate signals from their original process pathway into chemotaxis signaling.

PP_2969(1) The authors present a sequence alignment (Figure S5) that is the sole basis for their initial assignment of this ORF as a CsoR protein. There is a conservation of the primary coordinating ligands (highlighted with asterisks) known to be involved in Cu(I) binding to CsoR (ref 31). There are some key differences, though, in residues immediately adjacent to the conserved Cys (the preceding Ala, which is Tyr in the other sequences). The effect of these changes may be significant in a physiological context.

We constructed a point mutation in PP_2969 by replacing the Ala residue before the conserved Cys with a Tyr (CsoR_A39Y_) and then analyzed the effect of this mutation on CsoR. As shown in Author response image 1a, CsoR_A39Y_ showed similar promoter-binding ability as the wild-type CsoR and the presence of Cu^2+^ abolished the interaction between CsoR_A39Y_ and DNA, suggesting that the A39 residue in PP_2969 was not essential for the DNA-binding and Cu^2+^-binding abilities. Besides, CsoR_A39Y_ interacted with CheA as the wild-type CsoR did (Author response image 1b), indicating that the Ala39 residue was not required to interact with CheA.

The CsoR from *B. subtilis* has a Tyr before the conserved Cys, which is the same as other sequences, and the BTH result showed that interaction existed between CsoR and CheA from *B. subtilis* (Fig. 7 in the manuscript).

**Author response image 1. sa2fig1:** The effect of CsoR point mutation (CsoR_A39Y_) on the DNA-binding and Cu^2+^-binding abilities of CsoR. (a) Analysis for interactions between CsoR/CsoR_A39Y_ and *copA-I* promoter DNA using EMSA. The concentrations of CsoR/CsoR_A39Y_ and Cu^2+^ added in each lane are shown above the gel. Free DNA and protein-DNA complexes are indicated. (b) The interaction between CsoR/CsoR_A39Y_ and CheA was tested by BTH. Blue indicates protein-protein interaction in the colony after 60 h of incubation, while white indicates no protein-protein interaction. CK+ represents positive control, and CK- represents negative control.

(2) The gene immediately downstream of PP_2969 is homologous to *E. coli* RcnA, a demonstrated Ni/Co efflux protein, suggesting that P2969 may be Ni or Co responsive. Indeed PP_2970 has previously been reported as Ni/Co responsive (J. Bact 2009 doi:10.1128/JB.00465-09). The host cytosol plays a critical role in determining metal response, in addition to the protein, which can explain the divergence from the metal response expected from the alignment.

Correction: The gene immediately upstream (not downstream) of *PP_2969* (the ID is *PP_2968*, not *PP_2970*) is homologous to *E. coli* RcnA, a demonstrated Ni/Co efflux protein. The previous JBact study (J. Bact 2009 doi:10.1128/JB.00465-09) named PP_2968 as MrdH, and *mrdH* disruption led to sensitivity to cadmium, zinc, nickel, and cobalt, but not copper. Their results also revealed that MrdH was a broad-spectrum metal efflux transporter with a substrate range including Cd^2+^, Zn^2+^, and Ni^2+^. However, the role of MrdH in Cu^2+^ efflux was not tested. Commonly, metal efflux transporter has a broad substrate spectrum, allowing transporters to influence bacterial resistance to a variety of metals (Munkelt et al., J Bacteriol. 2004, 186:8036-8043; Grass et al., J Bacteriol. 2005, 187:1604-1611; Nies et al., J Ind Microbiol. 1995, 14:186-199; Kelley et al., Metallomics. 2021, 13:mfaa002). Our results showed that PP_2969 bound to Cu^2+^, Zn^2+^, and Ni^2+^ under our experimental conditions, and CsoR regulated the expression of genes related to Cu^2+^, Zn^2+^, and Ni^2+^ resistance, indicating that CsoR was involved in resistance to these metals. But the binding of CsoR to Cu^2+^ was the strongest, and Cu^2+^ showed the most substantial inhibitory effect on CsoR-DNA interaction. Thus, we considered its primary function as a Cu^2+^-responsive regulator.

(3) The previous JBact study also explains the lack of an effect (Figure 5b) of deleting PP_2969 on copper-efflux gene expression (copA-I, copA-II, and copB-II) as these are regulated by CueR not PP_2969 consistent with the previous report. Deletion of CsoR/RcnR family regulator will result in constitutive expression of the relevant efflux/detoxification gene, at a level generally equivalent to the de-repression observed in the presence of the signal.

We performed qPCR to test the effect of PP_2969 on gene expression, and we chose 14 target genes, including copper-resistance genes, nickel-resistance genes, zinc-resistance genes, cadmium-resistance genes, and cobalt-resistance genes. The results showed that *PP_2969* deletion led to a weak but significant increase (about 1.3-fold) in the expression of 10 genes (Fig. 4b, new Fig. S5a in the manuscript), and complementation with a multicopy plasmid containing *PP_2969* gene decreased the gene expression in Δ*PP_2969*. We were confused about these results. Why was the effect of PP_2969 on gene expression so weak? Did we pick the wrong target genes? In other bacteria, deletion of *csoR* led to an about ten-fold increase in gene expression, generally equivalent to the de-repression observed in the presence of metal. Thus, to further identify target genes, we performed RNA-seq to compare the gene expression in WT and Δ*PP_2969* without copper. The result surprised us because no gene expression levels changed more than two-fold (data not shown). This result might be attributed to several vital regulators that activated the expression of metal-resistance genes in response to metal in *P. putida*, such as CueR and CopR (Adaikkalam and Swarup, Microbiology. 2002, 148:2857-2867; Hofmann et al., Int J Mol Sci, 2021, 22:2050; Quintana et al., J Biol Chem, 2017, 292:15691-15704). CueR positively regulated the expression of *cueA*, encoding a copper-transporting P1-type ATPase that played a crucial role in copper resistance. CopR was essential for expressing several genes implicated in cytoplasmic copper homeostasis, such as *copA-II*, *copB-II*, and *cusA*. The existence of these positive regulators might make the function of CosR a secondary or even dispensable insurance in the expression of copper-resistance genes. Consistent with this, there is no CosR homolog in *P. aeruginosa*, and copper homeostasis is mainly controlled by CueR and CopR.

(4) Further, CsoR proteins are Cu(I) responsive so measuring Cu(II) binding affinity is not physiologically relevant (Figures 5a and S5b). The affinities of demonstrated CsoR proteins are 10-18 M and these values are determined by competition assay. The MTS assay and resulting affinities are not physiologically relevant.

Thank you for this enlightening comment. This question also confused us during our experiment. The first study on CsoR from *Mycobacterium tuberculosis* showed that CsoR bound a single-monomer mole equivalent of Cu(I) to form a trigonally coordinated complex, and that was a convincing result from protein structure analysis (Liu et al., Nat Chem Biol. 2007, 3:60-68). They further revealed that the presence of Cu(I) in the EMSA assay abolished the DNA-binding ability of CsoR, but the impact of Cu(II) was not tested. Besides, their results also showed that adding CuCl_2_ in the medium induced the expression of the *cso* operon involved in copper resistance. Perhaps Cu(II) converted to Cu(I) and then bound to CsoR in bacterial cells. Later studies in diverse bacterial species (including *Listeria monocytogenes*, *Corynebacterium glutamicum*, *Deinococcus radiodurans*, and *Thermus thermophilus*) showed that *in vitro* assays with Cu(II) abolished the DNA-binding ability of CsoR, indicating that CsoR bound to both Cu (I) and Cu(II) (Corbett et al., Mol Microbiol. 2011, 81:457-472; Teramoto et al., Biosci Biotechnol Biochem. 2012, 76:1952-1958; Zhao et al., Mol Biosyst. 2014, 10:2607-2616; Sakamoto et al., Microbiology. 2010, 156:1993-2005). Here, our results from *in vitro* assays (MST and EMSA) showed that CsoR bound to Cu(II) and Cu(II) affected the interaction between CsoR and promoter DNA. Compounds containing Cu(I) are poorly soluble in water and easily oxidized by Cu(II). DTT can reduce Cu(II) to Cu(I) (Krzel et al., J Inorg Biochem. 2001, 84:77-88). To test whether Cu(I) bound to CsoR and affected its DNA-binding ability, we recently performed an EMSA assay with the addition of CuCl_2_/DTT/CuCl_2_+DTT. As shown in Fig. 4d, the addition of DTT (0.1 and 1 mM) decreased CsoR-DNA interaction in the presence of 0.2 mM CuCl_2_, while the addition of DTT alone had no apparent influence on CsoR-DNA interaction, indicating that DTT enhanced the inhibition of CuCl_2_ on CsoR-DNA interaction, and the Cu(I) converted from Cu(II) by DTT had stronger inhibitory effect than Cu(II) on CsoR-DNA interaction. Together, these results suggested that CsoR bound to Cu(I) more strongly than it bound to Cu(II). We have added these results to the new version of manuscript.

(5) The DNA-binding assays are carried out at protein concentrations well above physiological ranges (Figures 5c and d, and S5c, d). The weak binding will in part result from using DNA sequences upstream of the copA genes and not from PP_2970.

We performed *the vitro* DNA-binding assay several times, and the lowest CsoR concentration used to obtain a shifted band was about 3 μM, and a higher concentration (15 μM) caused total DNA binding. Thus, we used the concentration of 15 and 20 μM to test the effect of metal on protein-DNA interaction in the assay. We also realized that these concentrations were above physiological ranges. We considered that the *in vitro* DNA-binding assay was only a mimic of the *in vivo* process, and the extracellular physiological conditions in EMSA might restrict the activity of CsoR. Besides, we recently performed EMSA to investigate the interaction between CsoR and its own promoter (*csoR*pro). As shown in Author response image 2, CsoR bound to *csoR*pro with a similar intensity to that it bound to *copA-I*pro. Thus, the weak binding was not caused by the promoter used in the assay.

**Author response image 2. sa2fig2:** The binding of CsoR to its own promoter (*csoR*pro) and *copA-I* promoter (*copA-1*pro) in EMSA. The concentrations of CsoR added in each lane are shown above the gel. Free DNA and CsoR-DNA complex are indicated.

CheA interactions(1) There is no consideration given to the likely physiological relevance of the new interacting partners for CheA.

Thank you for this comment. The initial purpose of this research was to identify new CheA-interacting proteins to broaden our knowledge of CheA and bacterial chemotaxis. Thus, we are currently focusing on the effect of the interaction on CheA and chemotaxis and trying to find the link between different processes and bacterial chemotaxis. We infer that the interaction between these new interacting partners and CheA can integrate signals from different pathways into the chemotaxis signaling pathway so that bacteria can better sense and adapt to different environments. Besides, the other role of the interaction, which is the influence of CheA on these new interacting partners, is also an exciting question that remains to be answered. Among the 16 new CheA-interacting proteins, five showed significant influence on chemotaxis, and the remaining 11 proteins had no obvious impact on chemotaxis (Fig. 2a in the manuscript). CsoR and PhaD inhibited CheA autophosphorylation, and here we focused on the effect of CsoR on chemotaxis. We also investigated the impact of CheA on CsoR, such as gene regulation and copper resistance. However, the results showed that CheA had no obvious influence on these functions of CsoR. The interactions between CheA and these proteins may be physiologically biased, with some interactions affecting the function of CheA and others mainly affecting the function of partners. Future studies on the function of these new CheA-interacting proteins and the role of CheA in regulating their functions would further expand our knowledge of CheA.

(2) How much CheA is present in the cell (copies) and how many copies of other proteins are present? How would specific responses involving individual interacting partners be possible with such a heterogenous pool of putative CheA-complexes in a cell? For PP_2969, the affinity reported (Figure 5A) may lay at the upper end of the CsoR concentration range (for example, CueR in *Salmonella* is present at ~40 nM).

Thank you for this insightful comment. We don’t know the copy number of CheA and other proteins in the cell. We were also initially surprised and felt skeptical about the reliability of CheA interaction with so many proteins. CheA interacts with CheY, CheW, and CheB in the classical chemotaxis pathway. This study found 16 new CheA-interacting proteins using pull-down assay and subsequent analysis. Moreover, in another unpublished result, we found that CheA interacted with eight c-di-GMP-metabolizing proteins, and CheA transferred the phosphate group to one of them. Together, it seemed that CheA could interact with at least 27 proteins. With such a heterogeneous pool of CheA-complexes, performing a specific response seemed difficult. However, several previous studies have reported the example of one protein interacting with dozens of proteins. For example, the c-di-GMP effector LapD in *Pseudomonas fluorescens* and *Pseudomonas putida* can interact with a dozen different c-di-GMP-metabolizing proteins (Giacalone et al., mBio. 2018, 9:e01254-18; Nie et al., Mol Microbiol. 2024, 121:1-17.) In *Escherichia coli*, a subset of DGCs and PDEs operated as central interaction hubs in a larger “supermodule” by interacting with dozens of proteins (Sarenko et al., mBio. 2017, 8:e01639-17). We infer that the expression of different CheA-interacting proteins might happen at different growth stages or under different conditions, and their interaction with CheA under that stage/condition changed bacterial chemotaxis or the process in which the target protein was involved.

(3) The two-hybrid system experiment uses a long growth time (60 h) before analysis. Even low LacZ activity levels will generate a blue color, depending upon growth medium (see doi: 10.1016/0076-6879(91)04011-c). It is also not clear how Miller units can be accurately or precisely determined from a solid plate assay (the reference cited describes a protocol for liquid culture).

We didn’t observe a blue color on the colony after 60 h growth on a plate under our experimental conditions. The BTH experiment was described as follows: After transforming the two plasmids into *E. coli* BTH101 cells, the plates containing transformants were placed at 28° for 48 h, at which time the colonies of the transformants were big enough to be picked up and incubated in a liquid medium for 24 h at 28°. Then, 5 μL of the culture was spotted onto an LB agar plate supplemented with antibiotics, X-gal, and IPTG and incubated for 60 h at 28° before taking the photos. After the photos were taken, the bacteria on the plate were scraped off and resuspended with buffer, and then the LacZ activity of the bacteria was tested. According to our experience, culture at 28°(lower than 30°) is a critical condition, and we have not observed false positives in BTH assays under this condition.

**Reviewer #1 (Recommendations For The Authors):**
In addition to genetic and biochemical approaches, structural studies should be conducted to elucidate the molecular interaction between CheA and CsoR with/without copper.It would be more logical to first establish the role of CsoR in copper regulation and chemotaxis (the second part of this report) and then investigate its underpinning mechanism (the first part).

Thank you for these recommendations. Structural analysis can reveal more details about the molecular mechanism of CheA-CsoR interaction, but we currently don’t have sufficient experimental conditions for such structural analysis.

As for the presentation logic of the results, we wrote the manuscript following the sequence of experiments. Firstly, screening of CheA interacting proteins (pull-down assay) was conducted, and then the influence of interacting proteins on the chemotaxis of strains and CheA autophosphorylation activity was detected. Based on these results, we obtained two proteins, CsoR and PhaD, and decided to go deeper into the function of CsoR and its effect on chemotaxis. We considered that this writing logic reflected our research design better and could also lay a foundation for future exploration of the functions of other interacting proteins and the physiological significance of interactions.

**Reviewer #2 (Recommendations For The Authors):**
A huge amount of effort has gone into this work.It would be good to see at least one of the newly identified interactions turn out to be physiologically relevant.The experimental tools appear to be available to do this, but it is critical to consider how these tools can lead to attempts to prove rather than test and possibly refute a model or hypothesis. In particular, please consider some of the comments about the physiological relevance of affinities when generating models.

Thank you for these recommendations. Our study aimed to screen new interacting proteins of CheA and explore how new interacting proteins affect CheA activity and bacterial chemotaxis, thereby broadening our understanding of chemotaxis. However, the impact of each protein-protein interaction has two sides: the influence of A to B and B to A. During experimental design, we focused more on the influence of identified interacting proteins on CheA function and chemotaxis but paid less attention to the function of interacting proteins and the influence of the interaction on their function. Moreover, our study found that the influence of protein-protein interaction was biased. In the interaction between CsoR and CheA, CsoR mainly affected the function of CheA and then affected the chemotaxis, while CheA had no significant effect on the function of CsoR. This might be attributed to the weak effect of CsoR in regulating metal resistance in *P. putida*, and we speculated that this interaction was more about favoring the sensing and avoiding metal stress. In addition, we planned to explore the interaction between CheA and another interacting protein (PhaD) in the future, reveal the effect of the interaction on PhaD function (regulation of PHAS synthesis in bacteria), and explore the effect of the interaction on CheA function and chemotaxis, to find out whether the association existed between PHAS anabolism and bacterial chemotaxis. Besides, for those proteins that did not have significant effects on CheA autophosphorylation and bacterial chemotaxis, we speculated that CheA might affect their function/activity through interactions, which meant that the physiological effects of the interaction mainly reflected through the interacting protein rather than CheA. These are speculations that need to be tested by experiments.